# Eelbrain, a Python toolkit for time-continuous analysis with temporal response functions

Christian Brodbeck[1]*, Proloy Das[2], Marlies Gillis[3], Joshua P Kulasingham[4], Shohini Bhattasali[5], Phoebe Gaston[1], Philip Resnik[6], Jonathan Z Simon[6]

[1]McMaster University, Hamilton, Canada; [2]Stanford University, Stanford, United States; [3]Katholieke Universiteit Leuven, Leuven, Belgium; [4]Linköping University, Linköping, Sweden; [5]University of Toronto, Toronto, Canada; [6]University of Maryland, College Park, College Park, United States

**Abstract** Even though human experience unfolds continuously in time, it is not strictly linear; instead, it entails cascading processes building hierarchical cognitive structures. For instance, during speech perception, humans transform a continuously varying acoustic signal into phonemes, words, and meaning, and these levels all have distinct but interdependent temporal structures. Time-lagged regression using *temporal response functions (TRFs)* has recently emerged as a promising tool for disentangling electrophysiological brain responses related to such complex models of perception. Here, we introduce the Eelbrain Python toolkit, which makes this kind of analysis easy and accessible. We demonstrate its use, using continuous speech as a sample paradigm, with a freely available EEG dataset of audiobook listening. A companion GitHub repository provides the complete source code for the analysis, from raw data to group-level statistics. More generally, we advocate a hypothesis-driven approach in which the experimenter specifies a hierarchy of time-continuous representations that are hypothesized to have contributed to brain responses, and uses those as predictor variables for the electrophysiological signal. This is analogous to a multiple regression problem, but with the addition of a time dimension. TRF analysis decomposes the brain signal into distinct responses associated with the different predictor variables by estimating a multivariate TRF (mTRF), quantifying the influence of each predictor on brain responses as a function of time(-lags). This allows asking two questions about the predictor variables: (1) Is there a significant neural representation corresponding to this predictor variable? And if so, (2) what are the temporal characteristics of the neural response associated with it? Thus, different predictor variables can be systematically combined and evaluated to jointly model neural processing at multiple hierarchical levels. We discuss applications of this approach, including the potential for linking algorithmic/representational theories at different cognitive levels to brain responses through computational models with appropriate linking hypotheses.

*For correspondence: brodbecc@mcmaster.ca

Competing interest: The authors declare that no competing interests exist.

## Editor's evaluation

Brodbeck et al. offer a timely and important contribution to how neural signals in response to continuous temporal modulations (as seen in speech and language processing) can be modelled effectively using temporal response functions. They offer a compelling new approach that includes a novel application of a boosting algorithm in addition to an accessible and didactically useful toolbox for analysis. A comparison of boosting and ridge regression via simulation shows the important impact on methods in speech and language neuroscience, as well as in cognitive neuroscience more broadly.

## Introduction

This paper introduces Eelbrain, a Python toolkit that makes it straightforward to express cognitive hypotheses as predictive computational models and evaluate those predictions against electrophysiological brain responses. The toolkit is based on the idea of decomposing brain signals into distinct responses associated with different predictor variables by estimating a multivariate temporal response function (mTRF), which maps those predictors to brain responses elicited by time-continuous stimulation (*Theunissen et al., 2001*; *Lalor et al., 2006*; *David et al., 2007*). This form of analysis has yielded valuable insights into the way that perception and cognitive processes unfold over time (e.g. *Ding and Simon, 2012*; *Broderick et al., 2018*; *Brodbeck et al., 2018a*; *Daube et al., 2019*; *Liberto et al., 2021*; *Sohoglu and Davis, 2020*).

### How to read this Paper

Time-lagged regression using TRFs is a mathematical method for analyzing the stimulus-response relationship between two signals that are evolving as a function of time, i.e., time series, like speech and brain activity measurements. The goal of this paper is to introduce several categories of cognitive neuroscience questions that can be asked using TRFs, and provide recipes for answering them. As such, the paper is not necessarily meant to be read in a linear fashion. The Introduction provides a general motivation for the approach and explains the underlying concepts in an accessible way. The *Results* section demonstrates how the technique can be applied to answer specific questions. The *Discussion* section highlights some more advanced considerations and caveats that should be kept in mind. The *Materials and methods* section explains the technical details and implementation in Eelbrain. The accompanying GitHub repository (*Brodbeck et al., 2023*) provides the source code for everything discussed in the paper (README.md contains instructions on how to get started). In addition, the Examples section on the Eelbrain website provides source code examples for many other basic tasks.

Depending on the background of the reader, these resources can be approached differently – for example, for readers new to mTRFs, we recommend reading the *Introduction* and *Results* sections first to get an idea of the questions that can be answered, and then referring to the Materials and methods for more detailed background information. On the other hand, readers specifically interested in the Eelbrain toolbox may want to skip ahead to the *Materials and methods* section early on.

### The convolution model for brain responses

The mTRF approach is built on the assumption that the brain response is continuously evolving in time as a function of the recently encountered stimulus (*Lalor et al., 2006*). Brain responses do not directly mirror a physical stimulus, but rather reflect a variety of transformations of that stimulus. For example, while speech is transmitted through air pressure variations in the kHz range, this signal is transformed by the auditory periphery, and macroscopic cortical responses are better described as responses to the slowly varying envelope of the original broadband signal. Thus, instead of directly predicting brain responses from the stimulus, the experimenter commonly selects one or several appropriate predictor variables to *represent* the stimulus, for example the low-frequency speech envelope (*Lalor and Foxe, 2010*).

The convolution model is a formal specification of how the stimulus, as characterized by the predictor variables, leads to the response. The stimulus-response relationship is modeled as a linear convolution in time, as illustrated in *Figure 1*. In contrast to classical analysis approaches that require averaging, the convolution model applies to single trial data and does not require any repetition of identical stimuli. A convolution kernel, or impulse response, characterizes the influence of each elementary unit in a predictor on the response. This kernel is also called the TRF, to distinguish it from the measured response to the stimulus as a whole. In addition to modeling an individual predictor variable (*Figure 1B, C*), the convolution model can also incorporate multiple predictor variables through the assumption that responses are additive (*Figure 1D*). Each predictor variable is associated with its own TRF, and thus predicts a separable response component. The ultimate response is the sum of those response components at each time point. This additive model is consistent with the fact that macroscopic measurements of electrical brain signals reflect an additive superposition of signals from different brain regions, potentially reflecting separable neural processes (*Nunez and Srinivasan, 2006*). When multiple predictor variables are jointly predicting a response, the collection

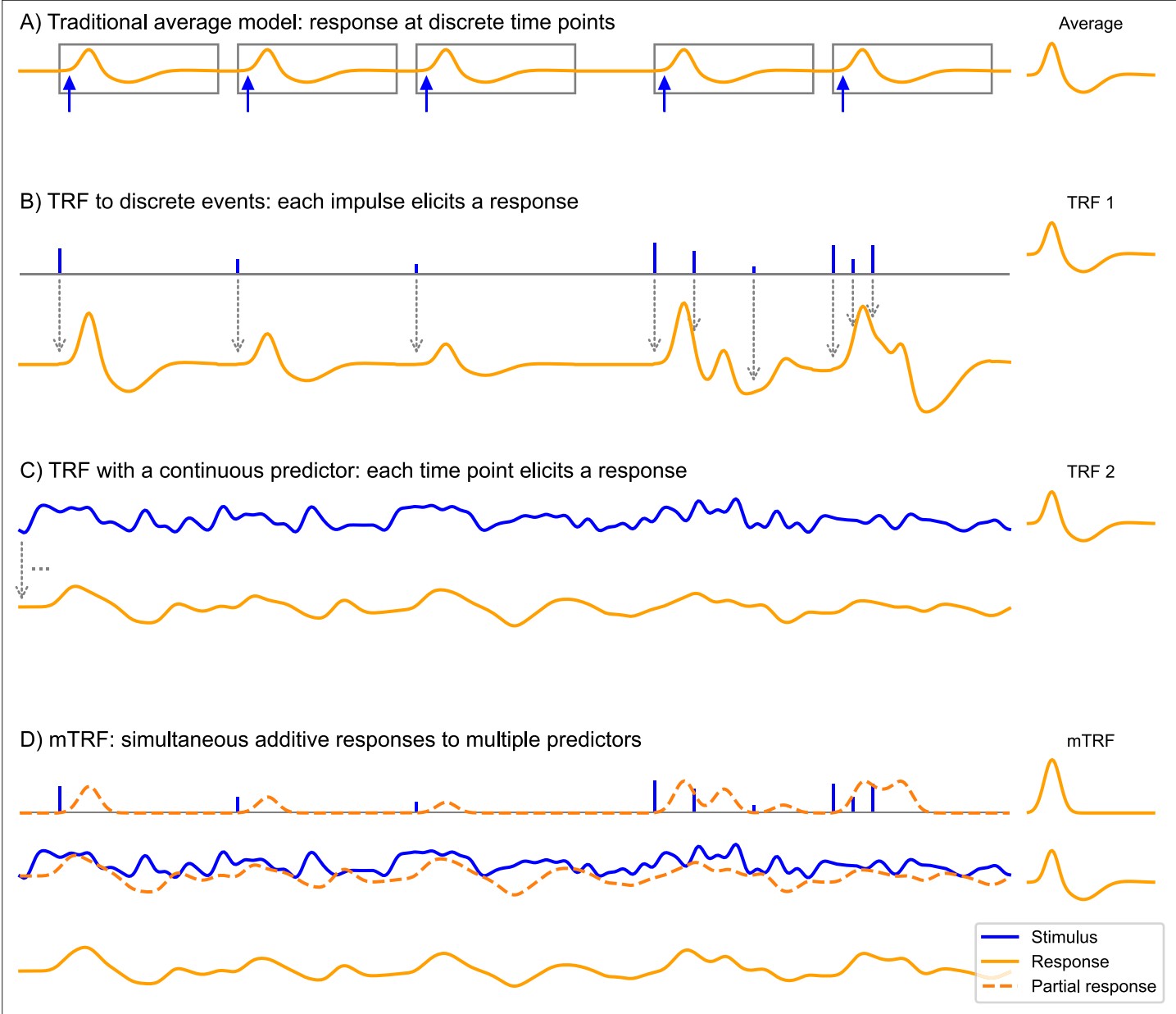

**Figure 1.** The convolution model for brain responses as a generalization of the averaging-based paradigm. (**A**) The traditional event-related analysis method assumes that each stimulus (blue arrows) evokes an identical, discrete response, and this response can be recovered by averaging. This model assumes that responses are clearly separated in time. (**B**) In contrast, the convolution model assumes that each time point in the stimulus could potentially evoke a response. This is implemented with time series predictor variables, illustrated here with a time series containing several impulses. These impulses represent discrete events in the stimulus that are associated with a response, for example, the occurrence of words. This predictor time series is convolved with some kernel characterizing the general shape of responses to this event type – the *temporal response function* (TRF), depicted on the right. Gray arrows illustrate the convolution, with each impulse producing a TRF-shaped contribution to the response. As can be seen, the size of the impulse determines the magnitude of the contribution to the response. This allows testing hypotheses about stimulus events that systematically differ in the magnitude of the responses they elicit, for example, that responses increase in magnitude the more surprising a word is. A major advantage over the traditional averaging model is that responses can overlap in time. (**C**) Rather than discrete impulses, the predictor variable in this example is a continuously varying time series. Such continuously varying predictor variables can represent dynamic properties of sensory input, for example: the acoustic envelope of the speech signal. The response is dependent on the stimulus in the same manner as in (**B**), but now every time point of the stimulus evokes its own response shaped like the TRF and scaled by the magnitude of the predictor. Responses are, therefore, heavily overlapping. (**D**) The multivariate TRF (mTRF) model is a generalization of the TRF model with multiple predictors: like in a multiple regression model, each time series predictor variable is convolved with its own corresponding TRF, resulting in multiple partial responses. These partial responses are summed to generate the actual complete response. Source code: figures/Convolution.py.

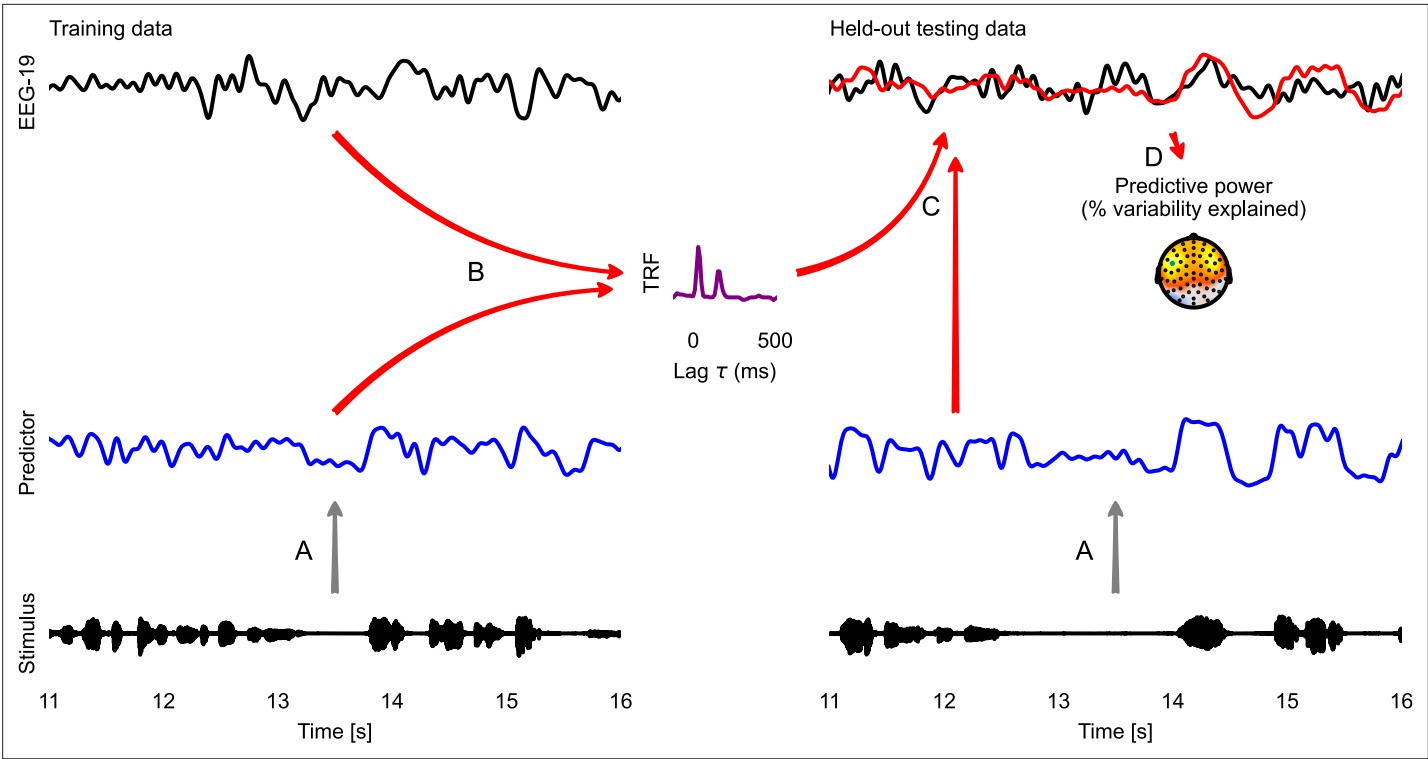

**Figure 2.** Temporal response function (TRF) analysis of EEG speech tracking. The left half illustrates the estimation of a TRF model, the right half the evaluation of this model with cross-validation. First, the stimulus is used to generate a predictor variable, here the acoustic envelope (**A**). The predictor and corresponding EEG data (here only one sensor is shown) are then used to estimate a TRF (**B**). This TRF is then convolved with the predictor for the held-out testing data to predict the neural response in the testing data (**C**; measured: black; predicted: red). This predicted response is compared with the actual, measured EEG response to evaluate the predictive power of the model (**D**). A topographic map shows the % of the variability in the EEG response that is explained by the TRF model, estimated independently at each sensor. This head-map illustrates how the predictive power of a predictor differs across the scalp, depending on which neural sources a specific site is sensitive to. The sensor whose data and TRF are shown is marked in green. Source code: figures/TRF.py.

of their TRFs is called an mTRF. As such, predictor variables can be thought of as hypotheses about how stimuli are represented by the brain, and multiple concurrent predictors can embody distinct hypotheses about how the stimulus is transformed across different brain regions (**Brodbeck and Simon, 2020**). The additive nature of the convolution model allows it to be applied to comparatively natural stimulus conditions, such as audiobook listening (**Hamilton and Huth, 2020**; **Alday, 2019**), while modeling natural variability through different predictor variables rather than minimizing it through experimental design.

In most practical data analysis scenarios, the true TRFs are unknown, but the stimulus and the brain responses are known. A TRF estimation algorithm addresses this, by estimating the mTRF that is optimal to predict the brain response from the predictor variables representing the stimulus. **Figure 2** illustrates this with EEG responses being predicted from the speech envelope. Typically, this is a very high-dimensional problem – including several predictor variables, each of which can influence the brain response at a range of latencies. Due to the large number of parameters, mTRFs are prone to overfitting, meaning that the mTRFs learn properties of the noise in the specific dataset rather than the underlying, generalizable responses. TRF estimation methods deal with this problem by employing different regularization schemes, i.e., by bringing additional assumptions to the problem that are designed to limit overfitting (see *Sparsity prior* below). A further step to avoid spurious results due to overfitting is evaluating model quality with cross-validation, i.e., evaluating the model on data that was never used during training. This step allows evaluating whether the mTRF model can generalize to *unseen* data and *predict* novel responses, as opposed to merely *explaining* the responses it was trained on.

## Nonlinear responses

Convolution can only model linear responses to a given input, whereas true neural responses are known to be nonlinear. Indeed, nonlinear transformations of the stimulus are arguably the most interesting, because they can show how the brain transforms and abstracts away from the stimulus, rather than merely mirroring it. We advocate a model-driven approach to study such nonlinear responses. A non-linear response can be modeled by generating a predictor variable that applies a non-linear transformation to the original stimulus, and then predicting brain responses as a linear response to this new predictor variable. For instance, it is known that the auditory cortex is disproportionately sensitive to acoustic onsets. This sensitivity has been described with a neural model of auditory edge detection, implemented as a signal processing routine (*Fishbach et al., 2001*). When this edge detection model is applied to the acoustic spectrogram, this results in a spectrogram of acoustic onsets, effectively mapping regions in the signal to which specific neuron populations should respond. This transformed spectrogram as a predictor variable thus operationalizes the hypothesis that neurons perform this non-linear transformation. Indeed, such acoustic onset spectrograms are highly significant predictors of auditory magnetoencephalography (MEG) responses (*Daube et al., 2019*; *Brodbeck et al., 2020*). Because mTRF models can only use linear transformations of the predictor variables to predict brain responses, a significant contribution from this predictor variable suggests that this non-linear transformation captures the non-linear nature of the neural processes giving rise to the brain responses.

This logic for studying nonlinear responses is taken to an even further level of abstraction when language models are used to predict brain responses. For instance, linguistic theory suggests that during speech comprehension, the continuous acoustic signal is transformed into discrete representations such as phonemes and words. However, we do not yet have an explicit, computational model of this transformation that could be used to generate an appropriate predictor. Instead, experimenters can estimate the result of an implicitly specified transformation based on extraneous knowledge, such as linguistic labels and corpus data. For example, responses to phonemes or phonetic features have been modeled through predictors reflecting discrete categories (*Di Liberto et al., 2015*). Furthermore, a series of such investigations suggests that brain responses to speech reflect linguistic representations at different hierarchical levels (*Brodbeck et al., 2018a*; *Broderick et al., 2018*; *Weissbart et al., 2020*; *Gillis et al., 2021*; *Brodbeck et al., 2022*). Such linguistic properties are commonly modeled as impulses corresponding to the onsets of words or phonemes. This does not necessarily entail the hypothesis that responses occur *at word onsets*. Rather, since the mTRFs allow responses at various latencies relative to the stimulus, such predictor variables can predict any time-locked responses that occur in an approximately fixed temporal relationship with the stimulus (within the pre-specified latency window).

During all this, it is important to keep in mind that even predictor variables that implement highly non-linear transformations are still likely to be correlated with the original stimulus (or linear transformations of it). For example, words and phonemes are associated with specific spectro-temporal acoustic patterns which systematically relate to their linguistic significance. Before drawing conclusions about non-linear transformations implemented by the brain, it is thus always important to control for more basic stimulus representations. In the domain of speech processing, this includes at least an acoustic spectrogram and an acoustic onset spectrogram (see Auditory response functions below). The latter in particular has been found to account for many responses that might otherwise be attributed to phonetic feature representations (*Daube et al., 2019*).

## This tutorial

For this tutorial, we use the openly available Alice dataset (*Bhattasali et al., 2020*) which contains EEG data from 33 participants who listened to the first chapter of *Alice in Wonderland* (12.4 min; 2129 words). The dataset also includes several word-level regressors derived from different syntactic language models, described in more detail in the original publication (*Brennan et al., 2019*). Here, we implement a complete group-level analysis for different levels of representation using Eelbrain.

Eelbrain implements mTRF estimation using boosting (*David et al., 2007*), as well as a variety of statistical tests, and ways to extract results for further analysis with other tools. The overall implementation of Eelbrain has been guided by the goal of facilitating mTRF estimation, group-level analysis, and visualization of results, for a general audience. The choice of boosting is significant as it encourages TRFs with a small number of non-zero coefficients, i.e., the boosting algorithm prefers a simpler

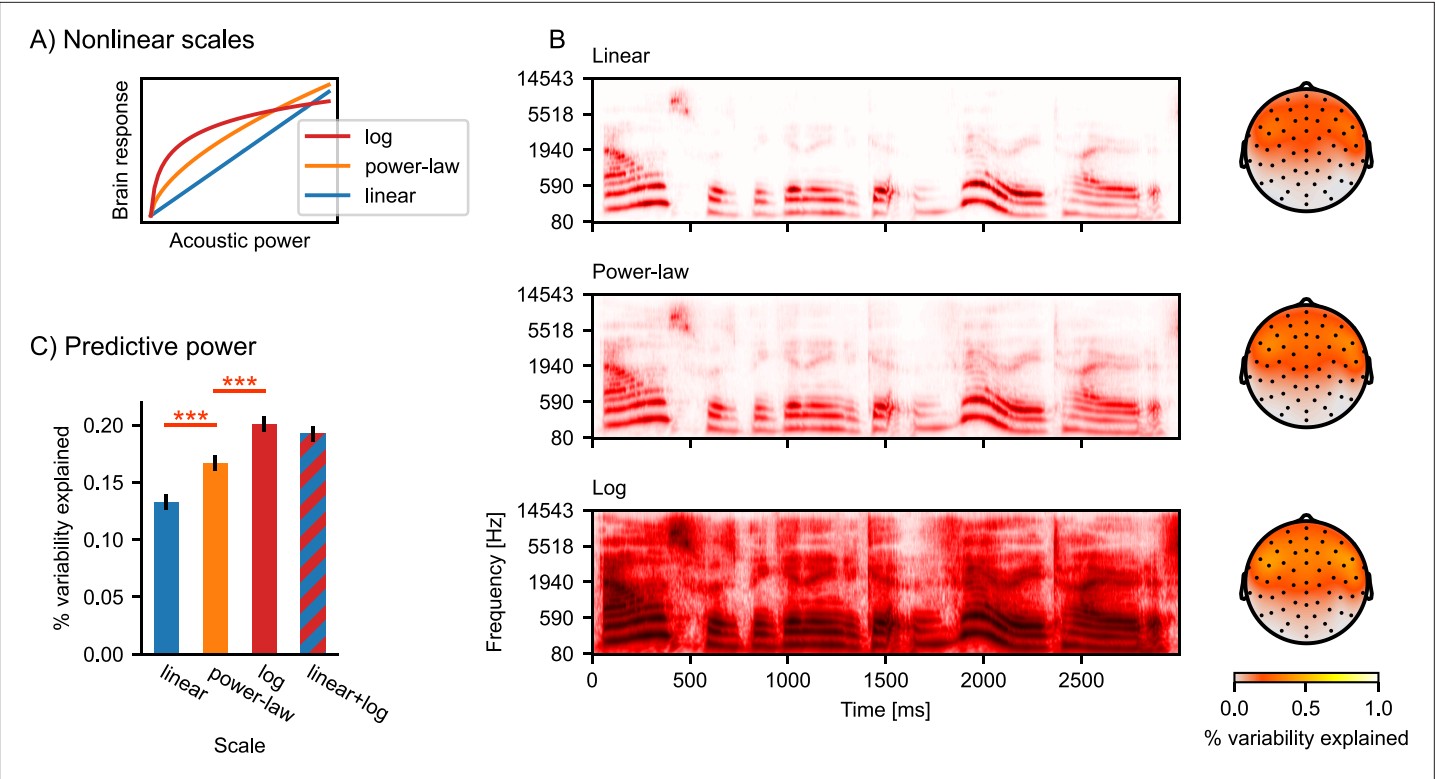

**Figure 3.** Nonlinear response scales. (**A**) Illustration of logarithmic and power-law scales, compared to linear scale. (**B**) Gammatone spectrograms were transformed to correspond to linear, power-law, and logarithmic response scales. Topographic maps show the predictive power of the three different spectrogram models. The color represents the percent of the variability in the EEG data that is explained by the respective multivariate TRF (mTRF) model. (**C**) Statistical comparison of the predictive power, averaged across all sensors. Error bars indicate the within-subject standard error of the mean, and significance is indicated for pairwise *t*-tests (*df* = 32). \*\*\*p≤0.001; Source code: figures/Auditory-scale.py.

explanation over a complex one (*Kulasingham and Simon, 2023*). This makes boosting suitable for estimating mTRFs for models consisting of structured, highly correlated, and possibly redundant predictor variables (*David and Shamma, 2013*), as is typical for models in cognitive neuroscience problems.

## Results

A critical goal of mTRF analysis for studying perception is evaluating brain responses that are nonlinear functions of the stimulus. In this section, we illustrate this by addressing increasingly complex nonlinear responses.

### Response scale

A basic nonlinearity is that of scale: brain responses can grow nonlinearly with the scale of the input. For instance, in the auditory system, neural responses tend to scale logarithmically with acoustic power encoded in a spectrogram (*Rahman et al., 2020*). This means that the same amount of power increase in the acoustic signal will cause a different increase in the brain response depending on the initial acoustic power value (see *Figure 3A*). Such nonlinear responses are typically modeled by scaling the predictor variable and assuming a linear relationship between the scaled predictor and response (e.g. *Fox, 2008*). Here, we determine the nonlinear relationship between acoustic power and brain response by comparing linear, logarithmic, and power law (*Biesmans et al., 2017*) scales.

*Figure 3B* shows the gammatone spectrogram, transformed to linear, power-law, and logarithmic response scales, along with the predictive power for EEG data resulting from the different transformations. When considering the average predictive power at all electrodes, the power-law scale spectrogram was a better predictor than the linear scale ($t(32) = 4.59$, p<0.001), and the log scale

further improved predictions compared to the power-law scale ($t(32) = 4.67$, p<0.001). Furthermore, nonlinear responses may mix properties of different scales. We thus tested whether the response may exhibit a linear component in addition to the logarithmic component by fitting a model including both the linear and the logarithmic spectrogram (i.e. twice as many predictors; similarly, polynomial regression combines multiple nonlinear transformations [*Fox, 2008*]). The linear + log combined model did not improve predictions over the logarithmic model (*Figure 3C*), suggesting that the EEG responses to the acoustic power are sufficiently described by the logarithmic response scale and do not contain an additional linear component.

## Auditory response functions

One goal of perception is to detect complex patterns present in the input signal, and we thus expect brain responses to represent features beyond simple acoustic intensity. Such features can be described as increasingly complex nonlinear transformations. Here, we illustrate this using acoustic onsets, a nonlinear transformation that is associated with strong responses and is a critical potential confound for higher-level features (*Daube et al., 2019*). Acoustic features of speech input are shown in *Figure 4A*: The upper panel shows the log-transformed gammatone spectrogram, quantifying acoustic energy as a function of time in different frequency bands. The gammatone filters simulate response characteristics of the peripheral auditory system. A simplified, one-dimensional representation of this spectrogram is the envelope, which is the summed energy across all frequency bands (blue line). The lower panel of *Figure 4A* shows an acoustic onset spectrogram based on a neurally inspired acoustic edge detection model (*Fishbach et al., 2001*), as described in *Brodbeck et al., 2020*. Again, a simplified one-dimensional version of this predictor, summing across all bands, signifies the presence of onsets across frequency bands (blue line).

To illustrate auditory features of increasing complexity we analyzed the following (m)TRF models:

$$acoustic\ envelope \tag{1}$$
$$acoustic\ envelope\ +\ acoustic\ onsets \tag{2}$$
$$acoustic\ spectrogram\ +\ onset\ spectrogram \tag{3}$$

*Figure 4B* shows response characteristics to the acoustic envelope alone. A topographic head-map shows the envelope's predictive power. The envelope alone is already a very good predictor of held-out EEG responses, with variability explained reaching 82% of that of the full spectro-temporal model (*Equation 3*) at anterior electrodes. The TRF to the envelope exhibits features characteristic of auditory evoked responses to simple acoustic stimuli such as tones or isolated syllables, including a P1-N1-P2 sequence.

*Figure 4C* shows the result of adding the one-dimensional acoustic onset predictor to the model. Together, onset and envelope significantly improve the prediction of held-out responses compared to just the envelope (p<0.001), indicating that the onset representation is able to predict some aspects of the EEG responses that the envelope alone cannot. The typical TRF to the onsets is of shorter duration than that to the envelope, and is characterized by two prominent peaks around 60 and 180 ms. The envelope TRF here is not much affected by adding the onset to the model (compare with *Figure 4B*).

*Figure 4D* shows the additional benefit of representing the envelope and onsets in different frequency bands, i.e., predicting EEG from an auditory spectrogram and an onset spectrogram (here eight bands were used in each, for a total of 16 predictors). As the predictors are two-dimensional (frequency × time), the resulting mTRFs are three-dimensional (frequency × lag × EEG sensor), posing a challenge for visualization on a two-dimensional page. One approach, assuming that response functions are similar across frequency bands, is to sum responses across frequency bands. As shown in *Figure 4D*, this indeed results in response functions that look very similar to the one-dimensional versions of the same predictors (*Figure 4C*). To visualize how the response functions differ for different frequency bands in the spectrograms, *Figure 4E* shows the full spectro-temporal response functions (STRFs), averaged across the electrodes that are most sensitive to the auditory stimulus features.

In sum, while the acoustic envelope is a powerful predictor of EEG responses to speech, additional acoustic features can improve predictions further, suggesting that they characterize neural representations that are not exhaustively described by the envelope. While the increase in prediction accuracy might seem small, more importantly, it allows the critical inference that the different predictors

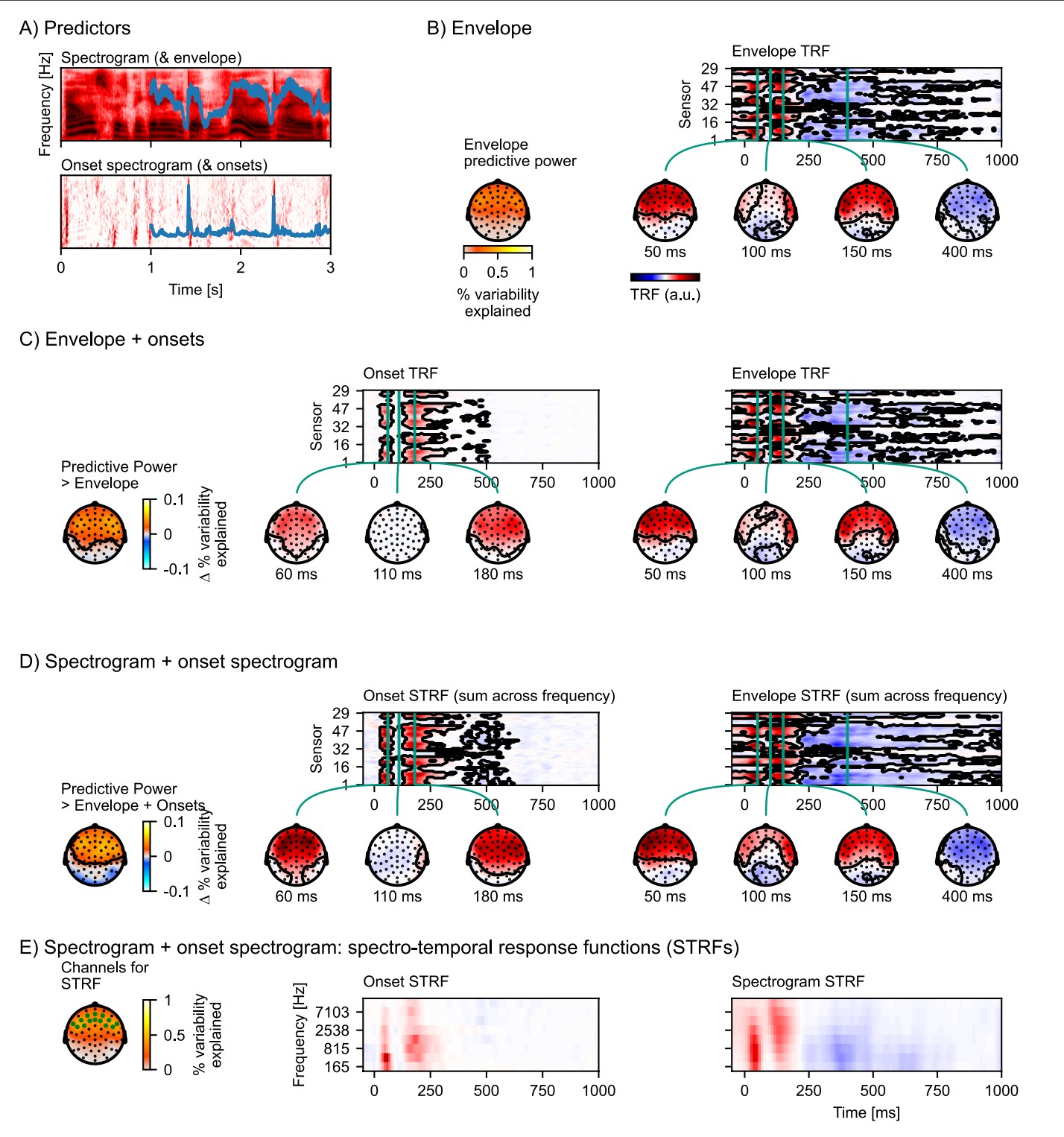

**Figure 4.** Auditory temporal response functions. (**A**) Common representations of auditory speech features: an auditory spectrogram (upper panel), and the acoustic envelope, tracking total acoustic energy across frequencies over time (blue line); and an acoustic onset spectrogram (lower panel), also with a one-dimensional summary characterizing the presence of acoustic onsets over time (blue line). (**B**) Brain responses are predicted from the acoustic envelope of speech alone. Left: Cross-validated predictive power is highly significant (p<0.001) at a large cluster covering all sensors. Right: the envelope temporal response function (TRF) – the y-axis represents the different EEG channels (in an arbitrary order), and the x-axis represents predictor-response time lags. The green vertical lines indicate specific (manually selected) time points of interest, for which head map topographies are shown. The black outlines mark significant clusters (p≤0.05, corrected for the whole TRF). For the boosting algorithm, predictors and responses are typically

*Figure 4 continued on next page*

*Figure 4 continued*

normalized, and the TRF is analyzed and displayed in this normalized scale. (**C**) Results for an multivariate TRF (mTRF) model including the acoustic envelope and acoustic onsets (blue lines in A). The left-most head map shows the percentage *increase* in predictive power over the TRF model using just the envelope (p<0.001; color represents the change in the percent variability explained; black outlines mark significant clusters, p≤0.05, family-wise error corrected for the whole head map). Details are analogous to (**B**). (**D**) Results for an mTRF model including spectrogram and onset spectrogram, further increasing predictive power over the one-dimensional envelope and onset model (p<0.001; color represents the change in percent variability explained). Since the resulting mTRFs distinguish between different frequencies in the stimulus, they are called spectro-temporal response functions (STRFs). In (**D**), these STRFs are visualized by summing across the different frequency bands. (**E**) To visualize the sensitivity of the STRFs to the different frequency bands, STRFs are instead averaged across sensors sensitive to the acoustic features. The relevant sensors are marked in the head map on the left, which also shows the predictive power of the full spectro-temporal model (color represents the percent variability explained). Because boosting generates sparse STRFs, especially when predictors are correlated, as are adjacent frequency bands in a spectrogram, STRFs were smoothed across frequency bands for visualization. a.u.: arbitrary units. Source code: figures/Auditory-TRFs.py.

are characterizing separable neural representations. For instance, a significant response to acoustic onsets (after controlling for the envelope/spectrogram) provides evidence for a nonlinear component of the brain response to speech that represents acoustic onsets. Separating the influence of different predictors is also important because different neural representations can have different response characteristics under different situations. For example, acoustic onsets might be especially important in segregating multiple auditory streams (*Brodbeck et al., 2020*; *Fiedler et al., 2019*).

## Word onsets as discrete events: Comparing ERPs and TRFs

While speech is continuous, perception is often characterized through discrete events. For example, in speech, words may be perceived as perceptual units. Such events have been analyzed using ERPs, but they can also be incorporated into an mTRF model, by using predictors that contain impulses at relevant time points (see *Figure 1*). In contrast to an ERP analysis, the mTRF analysis allows controlling for brain responses related to acoustic processing, and overlapping responses to events (words) that are close in time. Here, we directly compare these two analysis approaches.

Word onset TRFs controlling for acoustic processing were estimated using model *Equation 4*:

$$acoustic\ spectrogram\ +\ onset\ spectrogram\ +\ all\ words \tag{4}$$

These word onset TRFs are shown in *Figure 5* alongside the classical ERP. As expected from the earlier discussion, the TRF and ERP patterns are similar. The time and scalp regions where the two estimates of the brain response to words differ significantly are marked with gray bars (*Figure 5A*) and contours (*Figure 5B, C*; assessed via related-measures *t*-tests controlling for multiple comparisons). A prominent difference is that the ERP contains stronger activity in the baseline period, and a larger deflection at late lags (starting at 600 ms), with a topography similar to acoustic activity (compare with *Figure 4*), indicating artifactual leakage from acoustic responses. Additionally, there is a significant difference at around 200–450 ms which could be attributed to the temporal spread of the observed P2 peak in the ERPs, as the P2 peak in the TRF is temporally better defined (i.e. sharper) than in the ERPs.

Consistent with expectations, the ERPs overestimate the responses to words. The elevated baseline activity in particular suggests that the ERP is more prone to including brain activity that is not strictly a response to the stimulus feature, because the baseline period precedes the onset of the word. This is not surprising considering that the ERP is just an average of neural signals before and after word onsets, which includes, besides the responses to the words, neural responses to the acoustic signal, as well as activities that are not time-locked (including overlapping responses to the previous and next word, given that the average interval between word onsets in this stimulus set is only 334 ms). This highlights an advantage of the TRF paradigm over classical ERPs: the TRF enables characterization of only the neural response time-locked to the particular feature of interest, while explicitly controlling for signal variability due to other features of the stimuli.

## Categories of events: Function and content words

Given the powerful response to acoustic features of speech, it is important to take these responses into account when investigating linguistic representations. To illustrate the advantage of an mTRF analysis that can take into account the acoustic stimulus features, we revisit an old question: do brain responses differentiate between content words (roughly, words conveying a meaning) and function

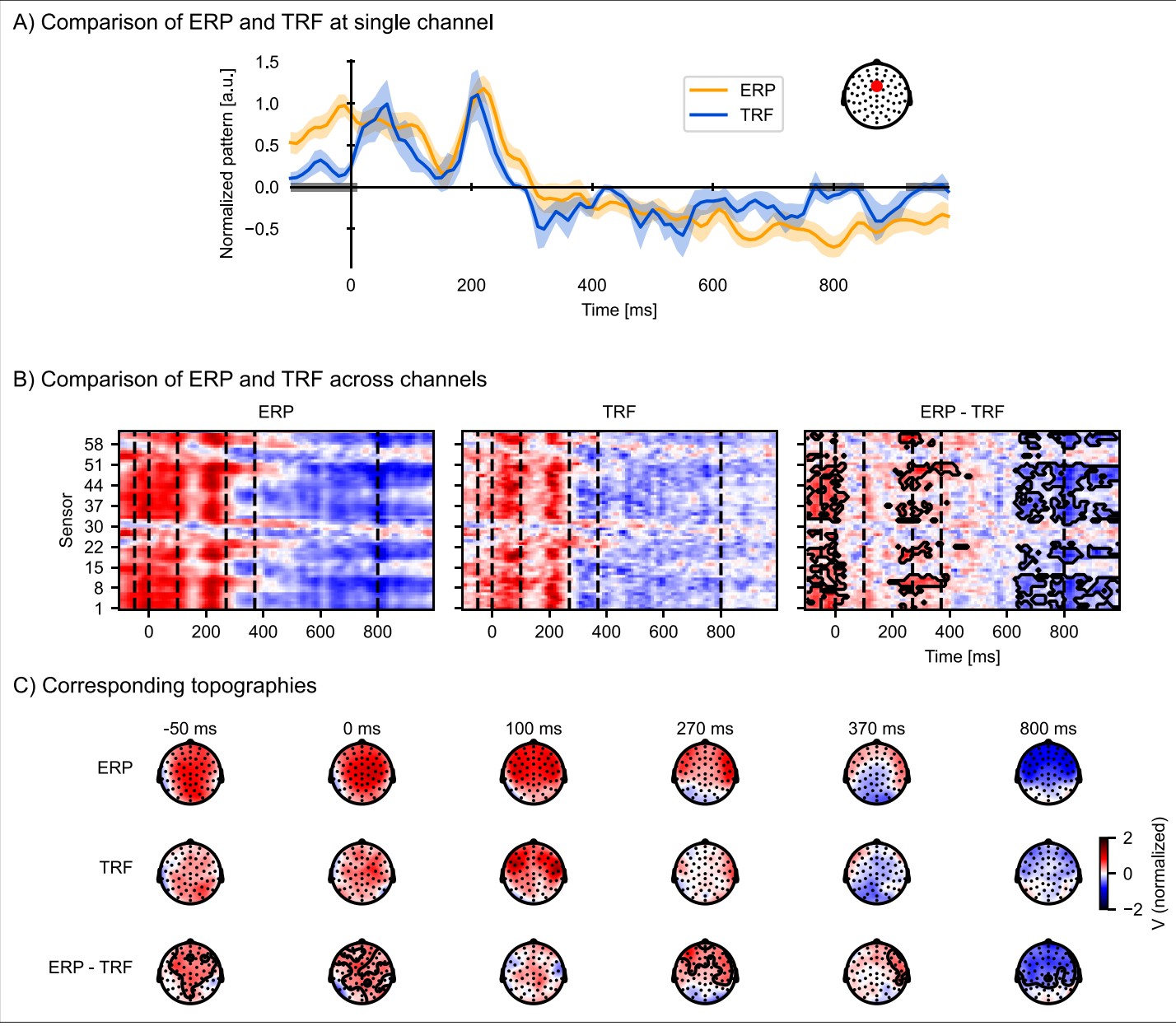

**Figure 5.** Temporal response functions (TRFs) to discrete events are similar to event-related potentials (ERPs), but control for acoustic processing and overlapping responses. To visualize ERP and TRF on a similar scale, both ERP and TRF were normalized (parametric normalization). (**A**) Visualization of the ERP and TRF response over time for a frontocentral channel, as indicated on the inset. The gray bars on the time axis indicate the temporal clusters in which the ERP and TRF differ significantly. Shading indicates within-subject standard errors ($n = 33$). (**B**) Visualization of ERP and TRF responses across all channels: the ERP responses to word onsets (left), the TRF to word onsets while controlling for acoustic processing (middle), and the difference between the ERP and the TRF (right). The black outline marks clusters in which the ERP and the TRF differ significantly in time and across sensors, assessed by a mass-univariate related-measures $t$-test. (**C**) Visualization of the topographies at selected time points, indicated by the vertical, dashed lines in (**B**), for respectively the ERP (top row), TRF (middle row), and their difference (bottom row). The contours mark regions where the ERP differs significantly from the TRF, as determined by the same mass-univariate related-measures $t$-test as in (**B**). a.u.: arbitrary units. Source code: figures/ Comparison-ERP-TRF.py.

words (words necessary for a grammatical construction)? In a first, naive approach, we ask literally whether brain responses differ between function and content words, while ignoring any potential confounding influences from acoustic differences. For this, we compare the predictive power of models *Equation 5* and *Equation 6*, all based on predictors with unit magnitude impulses at word onsets:

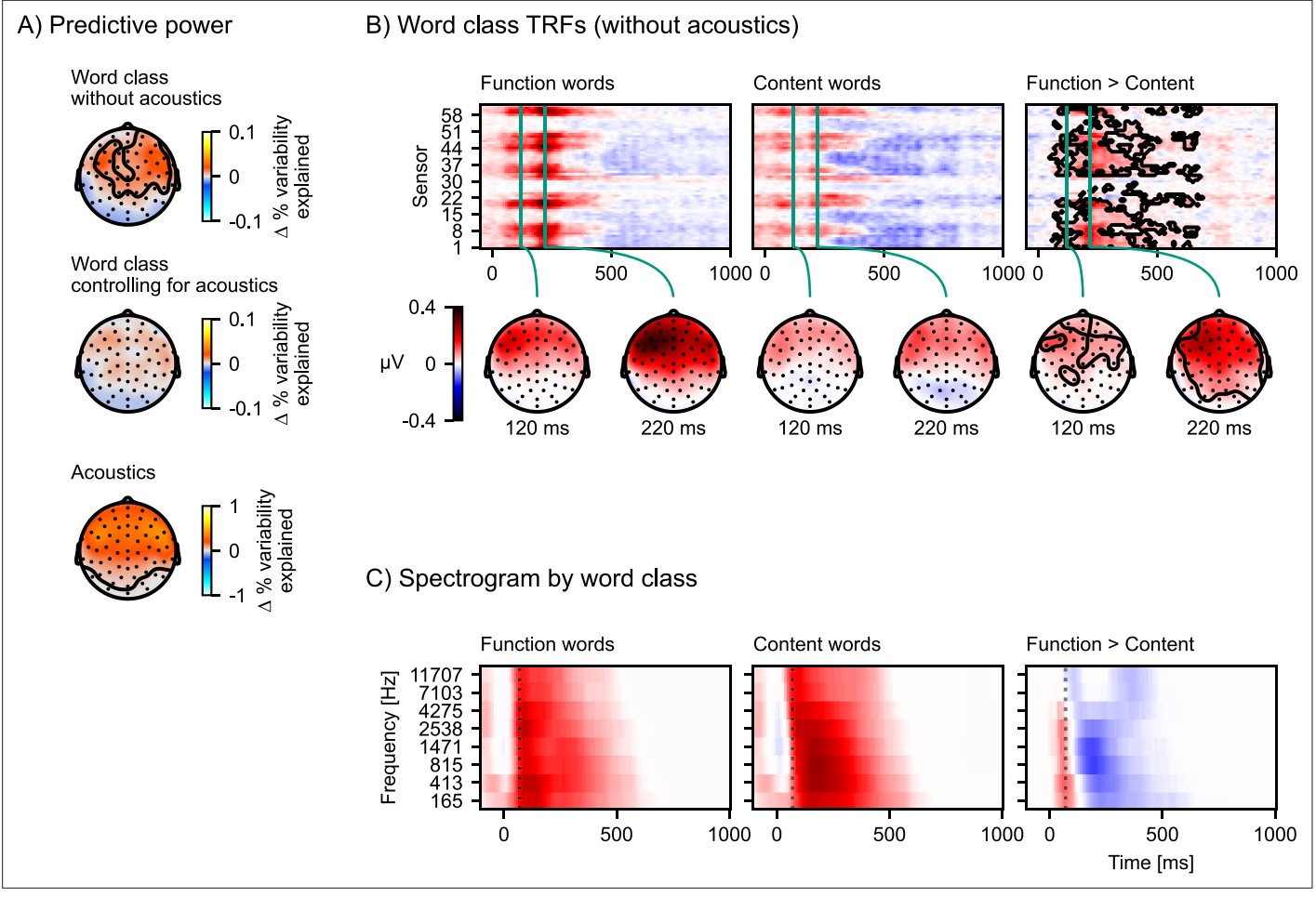

**Figure 6.** The difference in response to content and function words explained by acoustic differences. (**A**) Model comparisons of predictive power across the scalp. Each plot shows a head-map of the change in predictive power between different pairs of models. Top: *Equation 5* > *Equation 6*; middle: *Equation 7* > *Equation 4*; bottom: *Equation 4* > *Equation 6*. Color represents the difference in percent variability explained. (**B**) Brain responses occurring after function words differ from brain responses after content words. Responses were estimated from the temporal response functions (TRFs) of model *Equation 5* by adding the word-class-specific TRF to the *all words* TRF. The contours mark the regions that are significantly different between function and content words based on a mass-univariate related-measures *t*-test. (**C**) Function words are associated with a sharper acoustic onset than content words. The average spectrograms associated with function and content words were estimated with time-lagged regression, using the same algorithm also used for TRF estimation, but predicting the acoustic spectrogram from the function and content word predictors. A dotted line is plotted at 70 ms to help visual comparison. Color scale is normalized. Source code: figures/Word-class-acoustic.py.

$$All\ words\ + function\ words\ +\ content\ words \tag{5}$$
$$All\ words \tag{6}$$

Note that *all words* and the two word-class-based predictors are mathematically redundant because the *all words* vector equals the sum of the *function words* and *content words* vectors. However, because of the sparsity prior employed through the boosting algorithm and cross-validation, such redundant predictors can still improve a model – for example, if there is a shared response among all words, then attributing this response once as a TRF to *all words* is sparser than attributing it twice, once for *function words* and once for *content words* separately (see also *Sparsity prior* below).

The predictive power of the model with the word class distinction (*Equation 5*) is significantly higher compared to the model without it (*Equation 6*) at a right anterior electrode cluster (p=0.002, *Figure 6A*, top). To investigate why the word class distinction improves the model's predictive power, we compare the TRFs to function and content words (*Figure 6B*). Both TRFs are reconstructed from the relevant model components: Whenever a function word occurs in the stimulus, there will be both an impulse in the *all words*, and one in the *function words* predictor (and vice versa for content words).

Accordingly, the full EEG response occurring at the function word is split into two components, one in the *all words*, and one in the *function words* TRF. In order to visualize these full EEG responses, the displayed TRF for function words consists of the sum of the TRFs of the *all words* and the *function words* predictors, and the TRF for content words consists of the sum of the TRFs of *all words* and *content words*. This comparison suggests that function words are associated with a larger positive response at anterior sensors than content words.

To investigate whether this difference in responses might be due to acoustic confounds, we control for brain responses to acoustic features in both models and compare *Equation 7* with *Equation 4*, repeated here for convenience:

$$acoustic\ spectrogram\ +\ onset\ spectrogram\ +\ all\ words\ +\ functionwords\ +\ content\ words \quad (7)$$

In contrast to the comparison without acoustics, the comparison of the predictive power of *Equation 7* with *Equation 4* no longer indicates a significant difference (p=0.065, *Figure 6A*, middle). This suggests that the information in the acoustic predictors can explain the difference between brain responses to function and content words in model *Equation 5*. To directly characterize the influence of the acoustic features we plot the predictive power of the acoustic features by comparing models *Equation 4* and *Equation 6*. This comparison suggests that acoustic features are highly predictive of brain signals at anterior sensors (*Figure 6A*, bottom), encompassing the region in which the word class distinction originally showed an effect.

If the difference between brain responses to function and content words disappears when controlling for acoustic features, that suggests that acoustic features should differ between function and content words. We can assess this directly with another time-lagged regression model. First, we estimated filter kernels (analogous to the mTRFs) to predict the auditory spectrogram from models *Equation 6* and *Equation 5* (script: analysis/estimate_word_acoustics.py). To be able to statistically evaluate these results, we used 15-fold cross-validation and treated the predictive accuracy from each test fold as an independent estimate of predictive power (for a similar approach see *Etard et al., 2019*). With predictive power averaged across frequency bands, the model distinguishing function and content words is significantly better at predicting the spectrogram than the model treating all words equally ($t(14)$ = 13.49, p<0.001), suggesting that function and content words indeed differ acoustically. Finally, the filter kernels to function and content words from this analysis can be interpreted as average acoustic patterns corresponding to the two word-classes (*Figure 6C*). This comparison suggests that function words, on average, have a sharper acoustic onset. Since auditory cortex responses are known to be sensitive to acoustic onsets, a reasonable explanation for the difference in neural responses by word class, when not controlling for acoustic features, is that it reflects these sharper acoustic onsets of function words.

## Discussion

While the sections above provide recipes for various aspects of mTRF analysis, we use this section to discuss a number of advanced considerations and caveats, and possible extensions.

### TRF analysis vs. predictive power

In the various *Results* sections, we always tested the predictive power of a predictor variable before analyzing the corresponding TRF. This has a good reason: model comparisons based on predictive power are generally more conservative than comparisons based on TRF estimates. First, TRFs are directly estimated on the training data, and are still prone to some overfitting (although the boosting algorithm aims to minimize that using the validation step and early stopping). Second, as in conventional regression models, if two predictors are correlated, that means that they might share some of their predictive power. Model comparisons address this by testing for the *unique* predictive power of a variable, after controlling for all other variables, i.e., by testing for variability in the dependent measure that can *only* be explained by the predictor under investigation. The mTRFs (i.e. regression coefficients) cannot properly be disentangled in this way (*Freckleton, 2002*), and will usually divide the shared explanatory power amongst themselves. This means that mTRF estimates may always be contaminated by responses to correlated variables, especially when the correlations among predictor variables are high. This consideration highlights the importance of testing the predictive power of

individual model components before interpreting the corresponding TRFs to avoid spurious conclusions due to correlated predictors. In sum, a significant result in a model comparison provides strong evidence that a given predictor contributes unique predictive power when compared to the other predictors included in the model. In contrast, a significant effect in a TRF should always be interpreted with care, as it may also reflect the influence of other, correlated variables, even if those are included in the model.

TRF analysis may have an additional utility in diagnosing a special relationship between predictors. Assume that the brain represents a signal, say $x_t$, and this signal can be decomposed into $x_{1,t}$ and $x_{2,t}$ such that $x_t = x_{1,t} + x_{2,t}$. In a model using the predictors $x_{1,t}$ and $x_{2,t}$, both will contribute significantly. Thus, one might conclude that the brain represents two quantities separately, when in fact it represents only the average of them. However, because $x_t = x_{1,t} + x_{2,t}$, TRFs to $x_{1,t}$ and $x_{2,t}$ in the model including only these two predictors should look identical, thus providing a diagnostic for such a case. If the TRF shapes do not look identical, then that means that the properties in $x_{1,t}$ and $x_{2,t}$ are represented at different latencies, i.e., that they are separable.

## Sparsity prior

In its general formulation, mTRF analysis is a regression problem, albeit a high-dimensional one. Such high-dimensional analysis methods are almost always marred by overfitting. The presence of a large number of free parameters and correlated predictor variables makes the mTRFs prone to discovering noise patterns that are idiosyncratic to the particular dataset, and which do not generalize well to other datasets. Most regression analysis methods deal with this problem through regularization using a well-informed prior and some form of cross-validation. These regularization schemes vary considerably; some employ an explicitly formulated prior, e.g., the regularization in ridge regression (*Crosse et al., 2016a*); while others are defined on the algorithmic level, e.g., the early stopping criterion for boosting (see *Background: The boosting algorithm*). The implicit prior in the boosting algorithm promotes sparsity, i.e., it forces unimportant filter kernel coefficients to exactly 0. For problems with a large number of correlated predictors, the boosting algorithm may be preferable to other sparsity-enforcing algorithms such as LASSO (*Hastie et al., 2007*).

This sparsity prior has some consequences that might be counterintuitive at first. For example, in regression models, it is common to center predictors. This does not affect the explanatory power of individual regressors, because a shift in the mean of one predictor will simply lead to a corresponding shift in the coefficient for the intercept term. In contrast to this, a sparsity prior will favor a model with smaller coefficients. Consequently, an uncentered predictor that can explain responses with a small coefficient in the intercept term will be preferable over a centered predictor that requires a larger intercept coefficient.

Another consequence of the same preference for sparser models is that sometimes mathematically redundant predictors can improve the predictive power of a model. An example is that when splitting an impulse predictor into two categories, such as when dividing words into function and content words, the original *all words* predictor does not necessarily become redundant (see *Categories of events: Function and content words*). This can make model construction more complex, but it can also be informative, for example by showing whether there is a common response component to all words that can be learned better by including an *all words* predictor, in addition to word-class-specific responses.

## Discrete events and coding

In practice, the assumption of millisecond-precise time-locking to force-aligned features might seem hard to defend. For example, identification of clear word and phoneme boundaries is an artificial imposition, because the actual acoustic-phonetic features blend into each other due to co-articulation. An mTRF analysis only requires time-locking on average to produce some consistent responses. Nevertheless, more precise time estimates enhance the model's ability to isolate the response. In order to model cognitive events more precisely, one might thus want to consider alternative event locations, for example the words' uniqueness points instead of word onsets.

Impulse coding is not the only coding that is available. Impulses are based on the linking hypothesis of a fixed amount of response per impulse. An alternative linking hypothesis assumes that the brain response is increased for the duration of an event. This could be modeled using a step function

instead of impulses. Yet another type of hypothesis might concern a modulation of another predictor. For example, the hypothesis that the magnitude of the neural representation of the speech envelope is modulated by how linguistically informative a given segment is can be implemented by scaling the speech envelope according to each phoneme's linguistic surprisal (*Donhauser and Baillet, 2020*). A related set of questions concerns whether the same predictor is more or less powerful in different experimental conditions (*Sohoglu and Davis, 2020*).

## Source localization

EEG data is often analyzed in sensor space. However, because each sensor records a mixture of underlying neural (and artifactual) signals from different sources, sensor space data is inherently noisy. When data is recorded from a dense enough sensor array, neural source localization can be used to estimate a reconstruction of the sources contributing to the mixture signal, and assign the different brain responses to their cortical locations (*Nunez and Srinivasan, 2006*). Approaches combining source localization with mTRF analysis can differentiate responses related to processing of continuous stimuli anatomically as well as temporally, and thus provide a way to investigate hierarchical models of sensory and cognitive processing involving multiple anatomical regions. In many cases, source localization can also improve the signal-to-noise ratio of a specific response, because it acts as a spatial filter, aggregating information that is relevant for a given brain region across sensors, and suppressing signals whose likely origin is a different location.

A straight-forward extension of the approach described here is to apply a linear inverse solution to the continuous data, and apply mTRF analyses to the virtual current dipoles (*Brodbeck et al., 2018b*). For this purpose, Eelbrain contains functions that directly convert MNE-Python source estimate objects to `NDVars`. A more advanced approach is the Neuro-Current Response Function technique, which performs mTRF estimation and source localization jointly, in a unified estimation problem. This approach allows an mTRF model at each virtual current dipole, and estimates those mTRFs collectively to optimize the prediction of MEG measurements in sensor space (*Das et al., 2020*).

## Choosing the reference strategy

Since EEG data consists of voltage measurements, but only voltage differences are physically meaningful, the choice of referencing strategy matters. There are several common choices for a voltage reference for EEG data, including the average signal recorded at the mastoid electrodes, the central electrode Cz, and the average across all channels (also called the common average reference). Although the reference has minimal influence on the magnitude of the prediction accuracies averaged across channels or the latencies of the TRF peaks, it does have a substantial impact on the distribution across the EEG channels (*Figure 7*). This is because, depending on the chosen referencing strategy, the neural sources are projected differently across the scalp. Therefore, the activity of the neural dipoles of interest might be more (or less) prominent depending on the chosen referencing strategy.

## Further applications

While cortical processing of speech has been a primary application for mTRF analysis, the technique has potential applications in any domain where stimuli unfold in time, and has already been successfully applied to music perception (*Liberto et al., 2021*; *Leahy et al., 2021*), audiovisual speech perception (*Crosse et al., 2016b*), and subcortical auditory processing (*Maddox and Lee, 2018*). Furthermore, mTRF analysis as discussed here assumes that TRFs are static across time. However, this is not always a valid assumption. For example, in multi-talker speech, TRFs to speech features change as a function of whether the listener attends to the given speech stream or not (*Ding and Simon, 2012*; *Brodbeck et al., 2018a*; *Broderick et al., 2018*). Thus, moment-to-moment fluctuations in attention might be associated with corresponding changes in the TRFs. While modeling this is a highly complex problem, with even more degrees of freedom, some initial progress has been made toward estimating mTRF models with dynamic TRFs that can change over time (*Babadi et al., 2010*; *Miran et al., 2018*; *Presacco et al., 2019*).

## Conclusions

TRF analysis has several advantages over ERP-based methods. In the *Results* section we demonstrated, with several examples, how TRFs can be used to estimate the brain response associated with

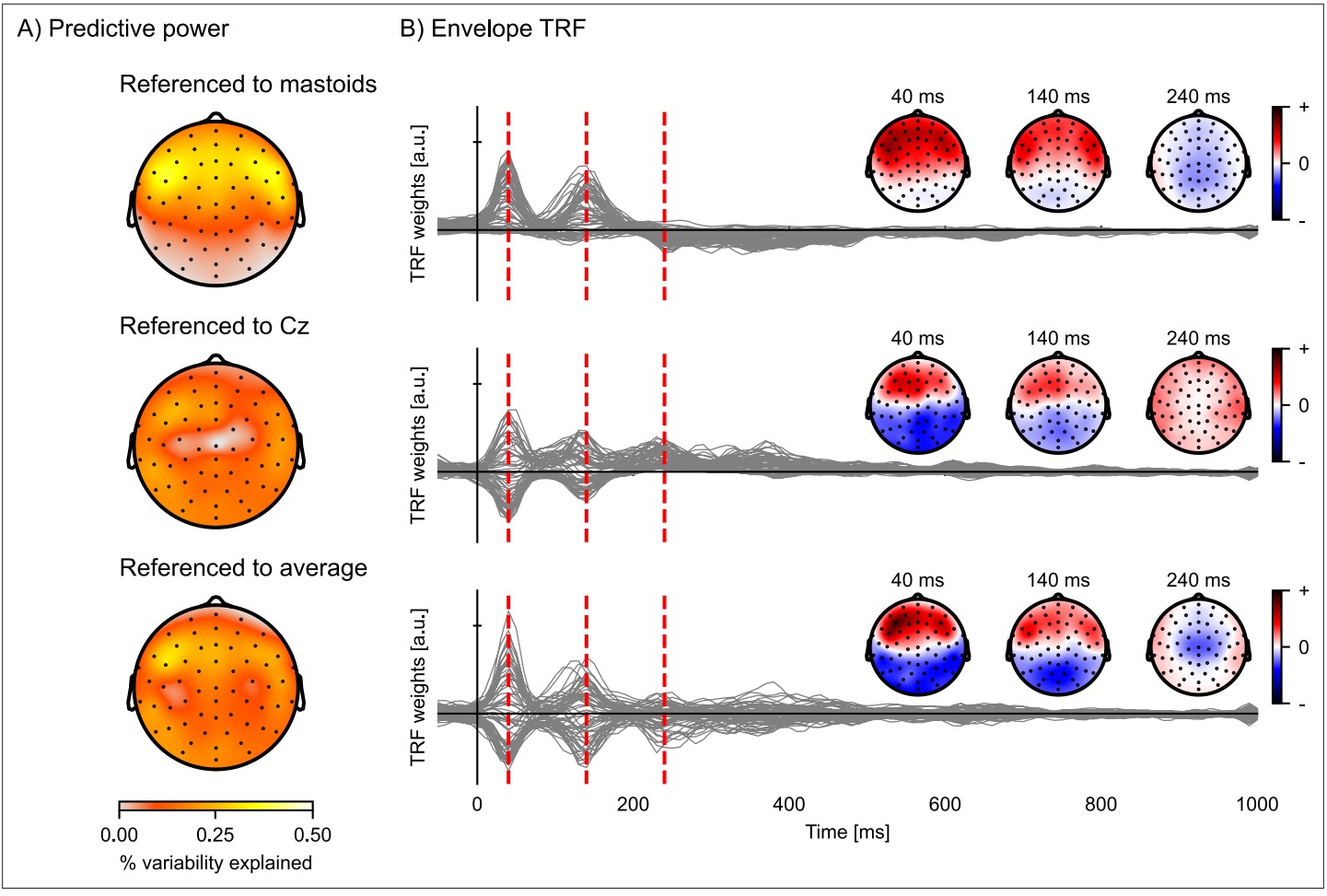

**Figure 7.** Comparison of EEG reference strategies. Predictive power topographies and temporal response functions (TRFs) to the acoustic envelope, according to three different referencing strategies: the average of the left and right mastoids (top), the central electrode Cz (middle), and the common average (bottom). (**A**) Visualization of the predictive power obtained with the different referencing strategies (color represents the percent variability explained). (**B**) The envelope TRFs for the different referencing strategies. The insets indicate the topographies corresponding to the vertical red dashed lines at latencies of 40, 140, and 240 ms. Source code: figures/Reference-strategy.py.

a given feature while controlling for (1) responses associated with other features that are correlated with the feature of interest, including correlations across different time lags (e.g. controlling for acoustics while analyzing responses to words), and (2) overlapping responses to events close in time (e.g. disentangling early responses to the current word from late responses to the previous word). In addition, analysis of the prediction accuracy allows for determining whether a given feature makes a unique contribution over and beyond other features of the same stimulus. This is especially important in naturalistic stimuli, where features are correlated in complex ways, because neural response estimates may be reliably different from zero even if the corresponding predictor does not provide unique predictive power. Finally, using cross-validation, models are assessed based on their *predictive* power, not just their *explanatory* power, thus providing an overall stronger test than conventional models.

## Materials and methods

This section describes each step towards a group-level analysis, starting from the data that is included in the open Alice EEG dataset (***Bhattasali et al., 2020***): EEG recordings, stimulus Wave (audio) files, and a comma-separated values (CSV) table with information about the words contained in the audiobook stimuli.

## Overall architecture of the Eelbrain toolbox

Eelbrain provides high-level functionality to represent time series data (such as EEG) and functions to work with this data. The `Dataset` class provides a data-table in which each row represents a measurement *case* (e.g. a trial), similar to a `dataframe` in R. The `Var` and `Factor` classes represent continuous and categorical columns in such a `Dataset`. In addition, the `NDVar` class (*n*-dimensional variable) provides a container for *n*-dimensional data. An `NDVar` instance also carries meta-information, for example the sampling rate of a time series, and sensor locations in EEG data. This allows other functions to access that information without user intervention. For example, a plotting function can directly generate topographic plots of an `NDVar` representing event-related potential data, without the user specifying which data point corresponds to which sensor, which dimension corresponds to the time axis, what the sampling rate is, etc. More detailed introductions can be found in the Examples section of the Eelbrain online documentation.

## Time series data

In the mTRF paradigm, a time series (for example, voltage at an EEG channel) is modeled as a linear function of one or several other time series (for example, the acoustic envelope of speech). The first step for an mTRF model is thus bringing different time-dependent variables into a common representational format. This is illustrated in *Figure 8*, which shows an excerpt from the first stimulus in the Alice dataset aligned to the EEG data from the first subject, along with different representations of the stimulus, which can model different neural representations.

As *Figure 8* suggests, time series will often have different dimensions. For example, EEG data might be two-dimensional with a time and a sensor dimension; a predictor might be one-dimensional, such as the acoustic envelope, or also have multiple dimensions, such as a spectrogram with time and frequency dimensions. To simplify working with different arbitrary dimensions, Eelbrain uses the `NDVar` (*n*-dimensional variable) class. An `NDVar` instance associates an *n*-dimensional numpy array (*Harris et al., 2020*) with *n* dimension descriptors. For example, the EEG measurements can be represented by a two-dimensional `NDVar` with two dimensions characterizing the EEG sensor layout (`Sensor`) and the time axis, as a uniform time series (`UTS`).

The first step for the mTRF analysis is thus to import the EEG measurements and predictor variables as `NDVar` objects, and align them on the time axis, i.e., make sure they are described by identical `UTS` dimensions. The `eelbrain.load` module provides functions for importing different formats directly, such as MNE-Python objects and Wave files (`NDVar` objects can also be constructed directly from numpy arrays). Keeping information about dimensions on the `NDVar` objects allows for concise and readable code for tasks such as aligning, plotting, etc. The code for *Figure 2* includes an example of loading a Wave file and aligning its time axis to the EEG data through filtering and resampling.

### EEG data

EEG data should be pre-processed according to common standards. In the Python ecosystem, MNE-Python offers a unified interface to a variety of EEG file formats and preprocessing routines (*Gramfort et al., 2014*). Here, we rely on the preprocessed data provided with the Alice EEG dataset, referenced to the averaged mastoids, and processed for artifact reduction with independent component analysis (see *Bhattasali et al., 2020*). However, a crucial additional step is filtering and downsampling the EEG data. When analyzing continuous electrophysiological recordings, removing low frequencies (i.e. high-pass filtering) takes the place of baseline correction, by removing high-amplitude slow drifts in the EEG signal which would otherwise overshadow effects of interest. Removing high frequencies, beyond the signals of interest (low-pass filtering), reduces noise and, more crucially, allows conducting the analysis at a lower sampling rate. This makes the analysis faster, because TRF estimation is computationally demanding, and processing times scale with the sampling rate. The major cortical phase-locked responses decrease quickly above 10 Hz (*Ding et al., 2014*), although there can be exceptions, such as a pitch-following response up to 100 Hz (*Kulasingham et al., 2020*). For the purpose of this tutorial, we are interested in the range of common cortical responses and apply a 0.5–20 Hz band-pass filter. Theoretically, a sampling rate exceeding two times the highest frequency (also known as Nyquist frequency) is necessary for a faithful representation of the signal. However, *Nunez and Srinivasan, 2006* recommend a sampling rate of 2.5 times the highest frequency due to various empirical considerations, such as the presence of random jitters, finite roll-off of the band-pass filter, etc. An

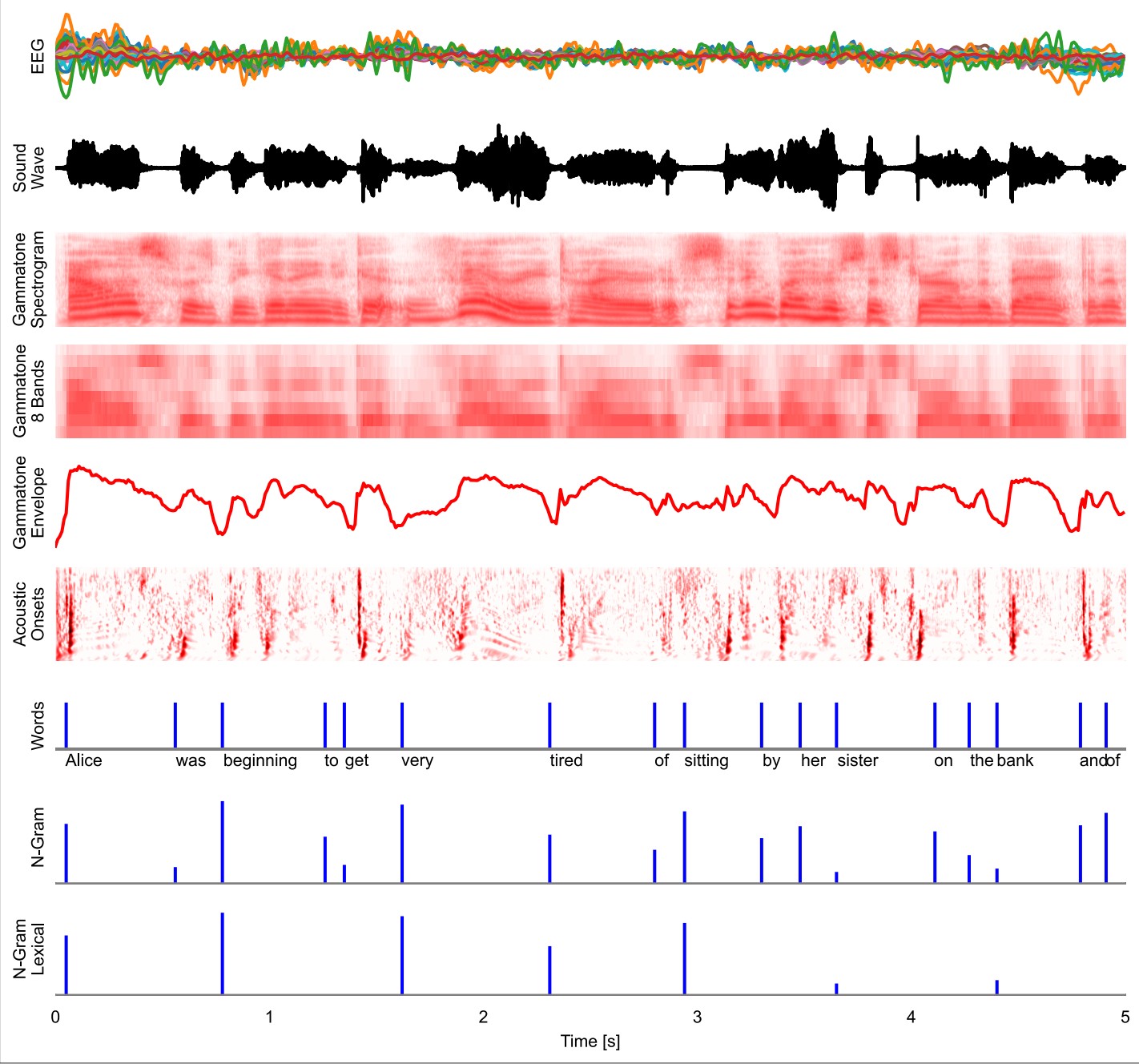

**Figure 8.** Time series representations of different commonly used speech representations, aligned with the EEG data. EEG: band-pass filtered EEG responses. Sound Wave: acoustic wave form, time-aligned to the EEG data. Gammatone Spectrogram: spectrogram representation modeling processing at the auditory periphery. Gammatone 8 Bands: the gammatone spectrogram binned into 8 equal-width frequency bins for computational efficiency. Gammatone Envelope: sum of the gammatone spectrogram across all frequency bands, reflecting the broadband acoustic envelope. Acoustic Onsets: acoustic onset spectrogram, a transformation of the gammatone spectrogram using a neurally inspired model of auditory edge detection. Words: a uniform impulse at each word onsets, predicting a constant response to all words. N-Gram: an impulse at each word onset, scaled with that word's surprisal, estimated from an *n*-gram language model. This predictor will predict brain responses to words that scale with how surprising each word is in its context. N-Gram Lexical: N-Gram surprisal only at content words, predicting a response that scales with surprisal and occurs at content words only. Source code: figures/Time-series.py.

even higher rate could be desirable for some secondary analysis and visualization, since it leads to smoother results. Here, we conduct the analysis with a sampling rate of 100 Hz.

EEG data usually contain markers that indicate the start of the stimulus presentation. These can be used to quickly extract EEG data time-locked to the stimuli in the required time series format, i.e., as a two-dimensional `NDVar` with `Sensor` and `UTS` dimensions (see source code to *Figure 2*).

## Predictor variables

Any hypothesis about time-locked neural processing that can be quantified can be represented by a time series derived from the stimulus. Here, we will illustrate two approaches: The first approach implements hypotheses about spectro-temporal transformations of the acoustic signal by directly applying those transformations to the speech waveform. The second approach implements hypotheses about linguistic processing based on experimenter-determined, discrete linguistic events.

### Time-continuous predictor variables: Gammatone spectrogram and derivatives

A common starting point for modeling acoustic responses is through a model of the cochlear transformation of the sound. Here, we use the gammatone spectrogram method to estimate cochlear transformations (*Patterson et al., 1992*; *Heeris, 2018*). A gammatone spectrogram is initially a high-dimensional representation, with more than a hundred time series representing the acoustic power at different frequency bands. For computational efficacy, the number of bands can be reduced by summarizing several contiguous bands into one. Here, we use eight bands as a compromise that leaves the global acoustic structure intact (*Figure 8*). An extreme form of this dimension reduction is using the acoustic envelope, which summarizes the entire spectrogram with a single band.

In addition to representing raw acoustic features, the auditory cortex is known to prominently represent acoustic onsets (*Daube et al., 2019*). Here, we model such representations by applying the neurally inspired auditory edge detection transformation to the gammatone spectrogram (*Brodbeck et al., 2020*). It is also common to approximate such a transformation through the half-wave rectified derivative of the acoustic envelope (*Fiedler et al., 2017*; *Daube et al., 2019*).

Because these predictor variables will be used repeatedly, it is convenient to generate them once and save them for future use. The script `predictors/make_gammatone.py` loops through all stimuli, computes a high-dimensional gammatone spectrogram (a two-dimensional `NDVar` with `frequency` and `UTS` dimensions), and saves it as a Python pickle file via the `eelbrain.save` module. The script `predictors/make_gammatone_predictors.py` loads these high-dimensional spectrograms via the `eelbrain.load` module and resamples them to serve as predictors, and it also applies the onset transformation and saves the resulting predictors.

Eelbrain provides functions to quickly generate this auditory model from audio files (`gammatone_bank` and `edge_detector`), as well as some basic signal processing routines for NDVars (e.g. `filter_data`; see the Reference section of the online documentation for a list). Different models can be constructed using different tools and converted to `NDVar`s (for an example of importing data from a `*.mat` file, see the EEG speech envelope TRF example of the online documentation).

### Discrete predictor variables

The analysis of linguistic representations commonly relies on forced alignment, a method that infers time-stamps for phoneme and word boundaries by comparing the sound waveform with a transcript. An example of an open-source forced aligner with extensive documentation is the Montreal Forced Aligner (*McAuliffe et al., 2017*). Because the forced alignment procedure requires some additional steps that are well documented by the respective aligners we skip it here, and instead use the word-onset time-stamps provided with the Alice dataset.

Discrete predictors come in two varieties: constant magnitude impulses and variable magnitude impulses (see *Figure 8*, lower half). Constant magnitude impulses always have the same magnitude, for example an impulse of magnitude 1 at each word onset. Such a predictor implements the hypothesis that all words are associated with a shared characteristic brain response, similar to an event-related potential (ERP). The TRF estimation algorithm will then determine the latencies relative to those impulses at which the brain exhibits a consistent response. Variable magnitude impulses implement the hypothesis that the brain response to each word varies systematically in amplitude with some

quantity. For example, the N400 amplitude is assumed to co-vary with how surprising the word is in its context. A predictor with an impulse at each word onset, whose magnitude is equal to the surprisal of that word, will enable predicting a stereotyped response to words whose amplitude linearly varies with word surprisal. The linking hypothesis here is that for each event, the brain responds with population activity that scales in amplitude with how surprising that event is, or how much new information it provides (see e.g. *Brodbeck et al., 2022*).

The Alice dataset comes with a table including all time-stamps and several linguistic properties (`stimuli/AliceChapterOne-EEG.csv`). Each row of this table represents one word, and contains the time point at which the word starts in the audio file, as well the surprisal values that were used to relate the EEG signal to several language models in the original analysis (*Brennan et al., 2019*). Such a table listing event times and corresponding feature values is sufficient for constructing appropriate regressors on the fly, and has a much lower memory footprint than a complete time series. The script `predictors/make_word_predictors.py` converts the table into the Eelbrain Dataset format that can be directly used to construct the regressor as an `NDVar`. To keep a common file structure scheme with the continuous regressors, such a table is generated for each stimulus.

Eelbrain provides a function for turning any time/value combination reflecting the occurrence of discrete events into a continuous predictor variable: `event_impulse_predictor`. The time/value pairs can either be constructed directly in Python, or can be imported, for example from a text file, using the `load.tsv` function.

## TRF estimation

## Background: The convolution model

The key assumption behind the mTRF approach is that the dependent variable, $y$, is the result of a convolution (linear filtering) of one or several predictor variables, $x$, with a corresponding filter kernel $h$. For a single predictor variable, the model is formulated as the convolution of the predictor variable with a one-dimensional filter kernel. For example, if $y_t$ is the value of an EEG channel at time $t$, and $x_t$ the value of the acoustic envelope at time $t$:

$$y_t = \sum_{\tau=\tau_{min}}^{\tau_{max}} h_\tau x_{t-\tau}$$

Here $h$ represents the filter kernel, also known as TRF, and $\tau$ enumerates the time delays or lags between $y$ and $x$ at which $x$ can influence $y$. To extend this approach to multiple predictor variables, it is assumed that the individual filter responses are additive. In that case, $x$ consists of $n$ predictor time series and thus has two dimensions, one being the time axis and the other reflecting $n$ different predictor variables. The corresponding mTRF is also two-dimensional, consisting of one TRF for each predictor variable:

$$y_t = \sum_{i=0}^{n} \sum_{\tau=\tau_{min}}^{\tau_{max}} h_{i,\tau} x_{i,t-\tau}$$

This model allows predicting a dependent variable $y$, given predictors $x$ and mTRF $h$.

In neural data analysis scenarios, typically the measured brain response and the stimuli are known, whereas the filter kernel $h$ is unknown. This leads to two reformulations of the general problem in which $x$ and $y$ are known, and $h$ is to be estimated; these represent alternative approaches for analysis. In the first, the so-called forward or encoding model, $h$ is optimized to predict brain responses from stimulus representations. In the second, the so-called backward, or decoding model, $h$ is optimized to reconstruct a stimulus representation from the neural measurements. The problems can both be expressed in the same general form and solved with the same algorithms. Eelbrain provides an implementation of the boosting algorithm (*David et al., 2007*), further described in Background: Boosting implementation below.

## Forward model (encoding)

Given a continuous measurement and one or more temporally aligned predictor variables, the reverse correlation problem consists of finding the filter kernel, or mTRF, that optimally predicts the response from the stimuli. The result of the convolution now becomes the predicted response:

$$\hat{y}_t = \sum_{i=0}^{n} \sum_{\tau=\tau_{min}}^{\tau_{max}} h_{i,\tau} x_{i,t-\tau} \tag{8}$$

The goal of the algorithm estimating the mTRF $h$ is to minimize the difference between the measured response $y_t$ and the predicted response $y_t$. **Figure 2** illustrates the estimation of a forward model for EEG data.

The `eelbrain.boosting` function provides a high-level interface for estimating mTRFs and returns a `BoostingResult` object with different attributes containing the mTRF and several model fit metrics for further analysis. Usually, for a forward model, the brain response is predicted from the predictors using positive (i.e. causal) lags. For example,

```
trf = boosting(eeg, envelope, 0, 0.500)
```

would estimate a TRF to predict EEG data from the acoustic envelope, with stimulus lags ranging from 0 to 500 ms (Eelbrain uses seconds as the default time unit). This means that an event at a specific time in the acoustic envelope could influence the EEG response in a window between 0 and 500 ms later. If the dependent variable has multiple measurements, for example multiple EEG channels, Eelbrain automatically assumes a mass-univariate approach and estimates a TRF for each channel. Negative lags, as in

```
trf = boosting(eeg, envelope, -0.100, 0.500)
```

are non-causal in the sense that they assume a brain response that precedes the stimulus event. Such estimates can nevertheless be useful for at least two reasons. First, if the stimulus genuinely represents information in time, then non-causal lags can be used as an estimate of the noise floor. As such they are analogous to the baseline in ERP analyses, i.e., they indicate how variable TRFs are at time points at which no true response is expected. Second, when predictor variables are experimenter-determined, the temporal precision of the predictor time series might often be reduced, and information in the acoustic speech signal might in fact precede the predictor variable. In such cases, the negative lags might be diagnostic. For example, force-aligned word and phoneme onsets assume the existence of strict boundaries in the speech signal, when in fact the speech signal can be highly predictive of future phonemes due to coarticulation (e.g. **Salverda et al., 2003**).

An advantage of the forward model is that it can combine information from multiple predictor variables. The boosting algorithm in particular is robust with a large number of (possibly correlated) predictor variables (e.g. **David and Shamma, 2013**). Eelbrain supports two ways to specify multiple predictor variables. The first is using a multi-dimensional predictor, for example a two-dimensional `NDVar`, representing a `spectrogram` with multiple `frequency` bands. These are used with the boosting function just like one-dimensional time series, and will be treated as multi-dimensional predictor variables, i.e., the different frequency bands will be jointly used for the optimal prediction of the dependent variable:

```
mtrf = boosting(eeg, spectrogram, 0, 0.500)
```

The second option for using multiple predictor variables is specifying them as a list of `NDVar` (one and/or two-dimensional), for example:

```
mtrf = boosting(eeg, [envelope, spectrogram], 0, 0.500)
```

## Backward model (decoding)

Instead of predicting the EEG measurement from the stimulus, the same algorithm can attempt to reconstruct the stimulus from the EEG response. The filter kernel is then also called a decoder. This can be expressed with **Equation 8**, but now $y_t$ refers to a stimulus variable, for example the speech envelope, and $x_{i,t}$ refers to the EEG measurement at sensor $i$ and time $t$. Accordingly, a backward model can be estimated with the same boosting function. For example,

```
decoder = boosting(envelope, eeg, -0.500, 0)
```

estimates a decoder model to reconstruct the envelope from the EEG data using a 500 ms window. Note the specification of delay values $\tau$ in the boosting function, from the point of view of the predictor variable: because each point in the EEG response reflects the stimulus preceding it, the delay values are negative, i.e., a given point in the EEG response should be used to reconstruct the envelope in the 500 ms window preceding the EEG response.

Because the EEG channels now function as the multivariate predictor variable, all EEG channels are used jointly to reconstruct the envelope. An advantage of the backward model is that it combines data from all EEG sensors to reconstruct a stimulus feature. It provides a powerful measure of how much information about the stimulus is contained in the brain responses, taken as a whole. A downside is that it does not provide a straight-forward way for distinguishing responses that are due to several, correlated predictor variables. For this reason, we will not further discuss it here. However, backward models have applications in other domains, where questions are not about specific representations, for instance attention decoding in auditory scenes (*O'Sullivan et al., 2015*).

## Background: The boosting algorithm

The general TRF estimation problem, i.e., finding the optimal filter kernels in forward and backward models, can be solved with different approaches. The Eelbrain toolkit implements the boosting algorithm, which is resilient to over-fitting and performs well with correlated predictor variables (*David et al., 2007*; *David and Shamma, 2013*). The boosting algorithm is intentionally designed with a bias to produce sparse mTRFs, i.e., mTRFs in which many elements are exactly zero. The algorithm thus incorporates a so-called sparsity prior, i.e., a prior to favor sparser models over alternatives (see *Sparsity prior* above).

Boosting starts by dividing the data into training and validation folds, and initializing an empty mTRF (all values $h_{i,\tau}$ are initially set to 0). It then iteratively uses the training data to find the element in the filter kernel which, when changed by a fixed small amount `delta`, leads to the largest error reduction. The following pseudo-code shows the general boosting algorithm:

```
mTRF[:] = 0
n_validation_error_increased = 0

while n_validation_error_increased < 2:
  current_training_error = error of mTRF in training set
  current_validation_error = error of mTRF in validation set

  # Find training error for all possible coordinate steps
  for element in mTRF:
    error_add[element] = error of mTRF with element = element + Δ
    error_sub[element] = error of mTRF with element = element - Δ

  # If no error reduction is possible, reduce Δ or stop
  if smallest error in error_add, error_sub > current_training_error:
   Δ *= 0.5
   if Δ ≥ mindelta:
     continue
   else:
     break

  # Update the mTRF and check validation set error
  mTRF = mTRF corresponding to smallest error in error_add, error_sub
  new_validation_error = error of mTRF in validation set
 if new_validation_error > current_validation_error:
  n_validation_error_increased +=
else:
```

```
n_validation_error_increased = 0

mTRF = the mTRF in the history with the smallest validation error
```

For multiple predictors, the search is performed over all the predictors as well as time lags, essentially letting the different predictors compete to explain the dependent variable. At each step, only a single element in the mTRF is modified. In that respect, boosting is a coordinate descent algorithm because it moves only at right angles in the parameter space. This promotes sparsity, as most elements stay at zero in every step. After each such delta change, the validation data is consulted to verify that the error is also reduced in the validation data. Once the error starts increasing in the validation data, the training stops. This early stopping strategy prevents the model from overfitting to the training data. In other words, the early stopping strategy in the boosting algorithm acts as an implicit prior to promote sparsity, i.e., it forces unimportant filter kernel coefficients to remain exactly 0.

The default implementation of the boosting algorithm constructs the kernel from impulses (each element $h_{i,\tau}$ is modified independently), which can lead to temporally discontinuous TRFs. In order to derive smoother TRFs, TRFs can be constructed from a basis of smooth window functions instead. In Eelbrain, the basis shape and window size are controlled by the `basis` and `basis_window` parameters in the boosting function. Their use is described in more detail in *Meta-parameter: Basis function* below.

Additionally, when using multiple predictors, it may be undesirable to stop the entire model training when a single predictor starts overfitting. In that case, the `selective_stopping` option allows freezing only the predictor which caused the overfitting, while training of the TRF components corresponding to the remaining predictors continues, until all predictors are frozen.

Finally, the default error metric for evaluating model quality is the widely used $\ell_2$ error. Due to squaring, however, the $\ell_2$ error is disproportionately sensitive to time points with large errors. In electrophysiology, large errors are typically produced by artifacts, and it is undesirable to give such artifacts a large influence on the model estimation. Since it is not trivial to exclude time intervals containing such artifacts from the analysis in continuous data, Eelbrain also allows the use of the $\ell_1$ error norm through the `error='l1'` argument, which improves robustness against such outlier data.

Usually, both the dependent variable and the predictor are centered around zero, and either standardized against the standard deviation (for $\ell_2$ error) or normalized by the $\ell_1$ norm (for $\ell_1$ error) before starting the boosting algorithm. The centering step is crucial to eliminate the intercept, since the boosting algorithm prefers sparse solutions, i.e., solutions with exact zero coefficients. When multiple predictors are used, this preprocessing step is applied separately to each of them, to ensure none of them affects the boosting procedure disproportionately. If required, this preprocessing step can be skipped through an argument to the boosting function: `scale_data=False`.

## Cross-validation

By default, the boosting function trains an mTRF model on all available data. Reserving some data for cross-validation, which allows model evaluation without overfitting, can be enabled by setting the test parameter to `test=True`. Since the boosting algorithm already divides the data into training and validation sets, enabling cross-validation entails splitting the data into three sets for each run: training, validation, and test sets. While the training and validation segments are consulted for estimating the TRFs (as described above), the test segment does not influence the estimation of the TRFs at all. Only once the TRF estimation is finalized, the final TRF is used to predict the responses in the test segment. To use the data optimally, Eelbrain automatically implements $k$-fold cross-validation, whereby the data is divided into $k$ partitions, and each partition serves as the test set once (see the Data partitions Eelbrain example). Thus, through $k$-fold cross-validation, the whole response time series can be predicted from unseen data. The proportion of the explained variability in this response constitutes an unbiased estimate of the predictive power of the given predictors. The `BoostingResult` object returned by the boosting function contains all these metrics as attributes for further analysis.

## Comparison to ridge regression

An alternative approach to boosting is ridge regression (*Crosse et al., 2021*), which uses Tikonov regularization. Tikonov regularization biases the mTRFs to suppress mTRFs in the stimulus subspace

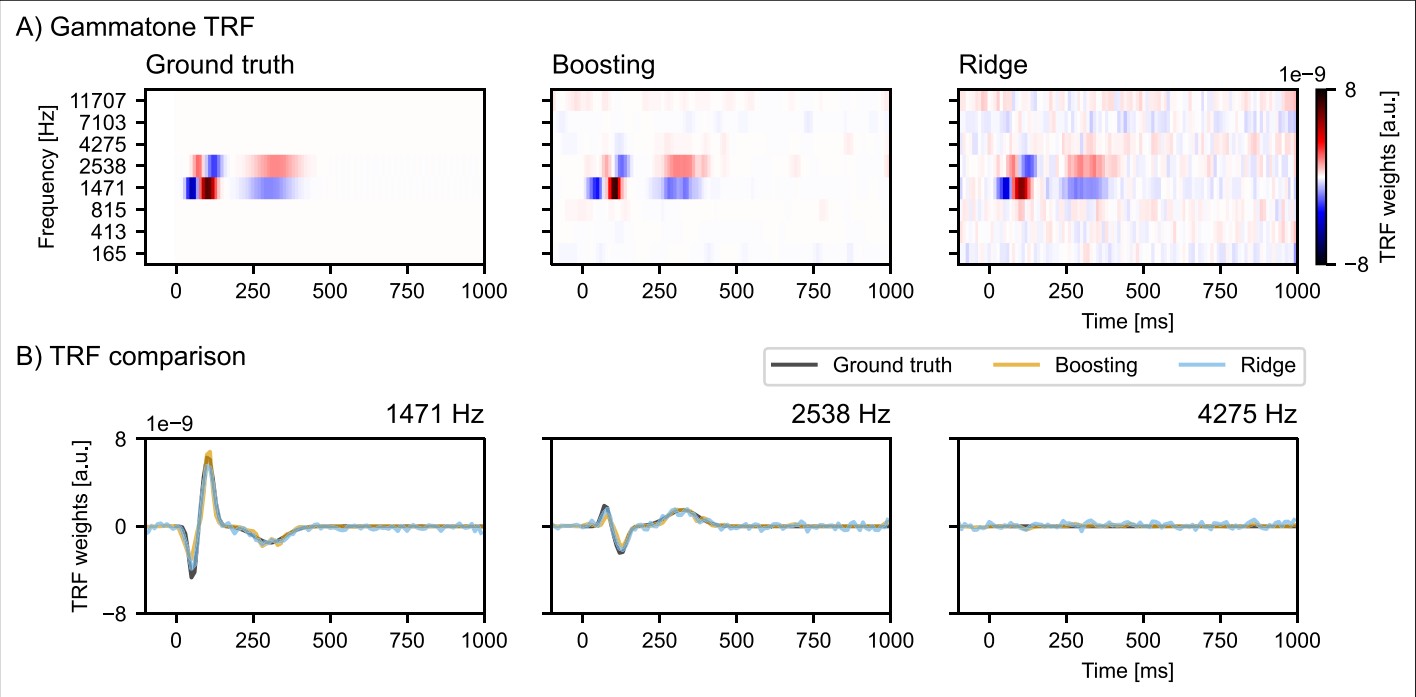

**Figure 9.** Simulation study comparing boosting with ridge regression when modeling multiple correlated variables. (**A**) A pre-defined multivariate temporal response functions (mTRF) was used to generate simulated EEG data (left panel), and then reconstructed by boosting and ridge regression (middle and right panel). (**B**) Overlays of boosting, ridge regression, and ground truth TRFs at different center frequencies for comparison. Note that the ridge TRF follows the ground truth closely, but produces many false positives. On the contrary, the boosting TRF enjoys an excellent true negative rate, at the expense of biasing TRF peaks and troughs toward 0. a.u.: arbitrary units. Source code: figures/Collinearity.py.

with a low signal-to-noise ratio, effectively imposing a smooth prior on the mTRF. In other words, Tikonov regularization tends to distribute the TRF power over all time lags. In contrast, boosting minimizes nonzero parameters in mTRFs, effectively imposing a sparseness prior on the mTRF, i.e., boosting concentrates the TRF power within a few time-lags. For a detailed discussion on the differences between boosting and ridge regression see *David et al., 2007*. *Figure 9* shows a side-by-side comparison between boosting and ridge regression in a simulation study where a gammatone spectrogram was used to simulate the time-locked EEG response.

Briefly, two adjacent gammatone bands (structured, highly correlated variables) were assumed to drive the auditory response with a spatiotemporally alternating pattern (*Figure 9A*, left panel). The simulated auditory response was then corrupted with additive pink noise to simulate EEG activity. We used the boosting function from Eelbrain and the ridge regression implementation from the pyEEG toolbox (*Weissbart et al., 2023*) to recover the mTRFs from the simulated EEG. For both algorithms, we used 10-fold cross-validation to select the optimum amount of regularization. With boosting we further tuned the `selective_stopping` hyper-parameter according to the explained variances in unseen data (with the `test = True` setting), i.e., we chose the model with the maximal `selective_stopping`, after which the explained variance started decreasing. For boosting, the TRFs are constructed from a basis of hamming windows with a window length of 50 ms, resulting in smoother TRFs. Both the mTRFs recovered by boosting and ridge regression capture the dominant features of the ground truth mTRF. However, the boosting mTRF has only a few peaks and troughs, giving a simpler representation with peaks and troughs closely corresponding to those of the ground truth, while the ridge regression mTRF includes many additional noisy peaks and troughs (*Figure 9A*, middle, and right panel; see also *David et al., 2007*; *David and Shamma, 2013*). In order to emphasize the differences between the estimates, the mTRF components are shown at the two Gammatone frequency bands driving the response (*Figure 9B*, left and middle panel), and at another frequency band with no response (rightmost panel). The ridge regression follows the dominant peaks and troughs of the ground truth mTRF closely, but, in doing so, acquires numerous false positives, i.e., small non-zero values when the true mTRF is actually zero. On the other hand, the boosting mTRF displays a high

true negative rate, while biasing the peaks and troughs towards zero. The two algorithms are further compared using multiple criteria in *Kulasingham and Simon, 2023*.

## Evaluating predictive power

In practice, a research question can often be operationalized by asking whether a specific predictor variable is neurally represented, i.e., whether it is a significant predictor of brain activity. However, different predictor variables of interest are very often correlated in naturalistic stimuli such as speech. It is thus important to test the explanatory power of a given variable while controlling for the effect of other, correlated variables. A powerful method for this is comparing the predictive power of minimally differing sets of predictor variables using cross-validation.

In this context, we use the term 'model' to refer to a set of predictor variables, and 'model comparison' refers to the practice of comparing the predictive power of two models on held-out data. It is important to re-estimate the mTRFs for each model under investigation to determine the effective predictive power of that model, because mTRFs are sensitive to correlation between predictors and can thus change depending on what other predictors are included during estimation.

When building models for a specific model comparison, we recommend a hierarchical approach: both models should include lower-level properties that the experimenter wants to control for, while the models should differ only in the feature of interest. For example, to investigate whether words are associated with a significant response after controlling for acoustic representations, one could compare the explained variability of *Equation 9* and *Equation 10*:

$$gammatone\ spectrogram\ +\ onset\ spectrogram \qquad (9)$$

$$gammatone\ spectrogram\ +\ onset\ spectrogram\ +\ word\ onset\ impulses \qquad (10)$$

If the model in *Equation 10* is able to predict held-out test data better than that in *Equation 9*, then this difference can be attributed to a predictive power of word onset impulses over and above the spectrogram-based representations.

Estimating such mTRF models is the computationally most demanding part of this analysis. For this reason, it usually makes sense to store the result of the individual estimates. The script `analysis/estimate_trfs.py` implements this, by looping through all subjects, fitting mTRFs for multiple models, and saving the results for future analysis.

## Group analysis

In order to statistically answer questions about the predictive power of different models we will need to combine the data from different measurements, usually from different subjects. To combine data from multiple subjects along with meta-information such as subject and condition labels, Eelbrain provides the `Dataset` class, analogous to data-table representations in other statistics programs, such as a `dataframe` in R or Pandas, but with the added capability of handling data from `NDVars` with arbitrary dimensionality.

A standard way of constructing a `Dataset` is collecting the individual cases, or rows, of the desired data table, and then combining them. The following short script provides a template for assembling a table of model predictive power for several subjects and two models (assuming the mTRF models have been estimated and saved accordingly):

```
cases = []
for subject in ['1', '2', '3']:
    for model in ['sgrams', 'sgrams+words']:
        mtrf = load.unpickle(f"path/to/{subject}_{model}.pickle")
cases.append([subject, model, mtrf.proportion_explained])

column_names = ['subject', 'model', 'explained']
data = Dataset.from_caselist(column_names, cases)
```

Thus, even though the `proportion_explained` attribute might contain 64 EEG channels (i.e. an `NDVar` with `Sensor` dimension) it can be handled as a single entry in this data table.

## Statistical tests

Statistical analysis of mTRFs faces the issue of multiple comparisons common in neuroimaging (*Maris and Oostenveld, 2007*). One way around this is to derive a univariate outcome measure. The (average) predictive power across all sensors is a measure of the overall predictive power of a model. If a prior hypothesis about the location of an effect is available, then the average predictive power at a pre-specified group of sensors can serve as a more targeted measure, for example:

```
sensors = ['45', '34', '35']
data['explained_average'] = data['explained'].mean(sensor = sensors)
```

Eelbrain implements a limited number of basic univariate statistical tests in the `test` module, but more advanced statistical analysis can be performed after exporting the data into other libraries. All univariate entries in a `Dataset` can be transferred to a `pandas.DataFrame` (*Reback et al., 2021*) with

```
dataframe = data.as_dataframe()
```

for analysis with other Python libraries like Pingouin (*Vallat, 2018*), or saved as a text file with

```
data.save_txt('data.txt')
```

to be transferred to another statistics environment like R (*R Development Core Team, 2021*).

Instead of restricting the analysis to a priori sensor groups, Eelbrain implements several mass-univariate tests (*Nichols and Holmes, 2002*; *Maris and Oostenveld, 2007*; *Smith and Nichols, 2009*). These tests, implemented in the `eelbrain.testnd` module (for *n*-dimensional tests), are generally based on calculating a univariate statistic at each outcome measure (for example, a *t* value corresponding to a repeated-measures *t*-test comparing the predictive power of two models at each EEG sensor), and then using a permutation-based approach for estimating a null distribution for calculating *p*-values that control for family-wise error. In Eelbrain, these tests can be applied to `NDVars` similarly to univariate tests, with additional arguments for controlling multiple comparison corrections. The code corresponding to *Figure 4* demonstrates a complete group analysis pipeline, from loading pickled mTRF models into a `Dataset` to plotting statistical results.

Mass-univariate tests can provide a detailed characterization of a given comparison, but calculating effect sizes for such results is not straightforward. For power analyses, carefully selected univariate outcome variables are usually more effective.

## TRF analysis

While predictive power is the primary measure to assess the quality of a model, the TRFs themselves also provide information about the temporal relationship between the brain response and the stimuli. A TRF is an estimate of the brain's response to an impulse stimulus of magnitude 1 (i.e. the impulse response). It thus characterizes the brain response as a function of time, relative to stimulus events, analogous to an ERP to a simple stimulus. For example, the TRF estimated to an acoustic envelope representation of speech commonly resembles the ERP to simple tone stimuli. The mTRFs can thus be analyzed in ways that are analogous to ERPs, for example using mass-univariate tests or based on component latencies. *Figure 4* demonstrates a representative analysis of TRFs and mTRFs corresponding to different auditory features.

## Meta-parameter: Basis function

By default, the fitting algorithm controls each element of a TRF independently of other elements. When a basis function is chosen, the algorithm instead generates the TRF by combining copies of that basis function, one centered on each element of the TRF. There are at least two reasons for selecting a basis function as part of the boosting algorithm for TRF estimation. First, constraining the TRFs to neurally more realistic shapes may increase the predictive power of the TRF model. Second, the resulting TRFs are smoother and thus easier to interpret and compare across subjects. *Figure 9* illustrates the effect of using a basis function, comparing a 50 or 100 ms wide Hamming window basis

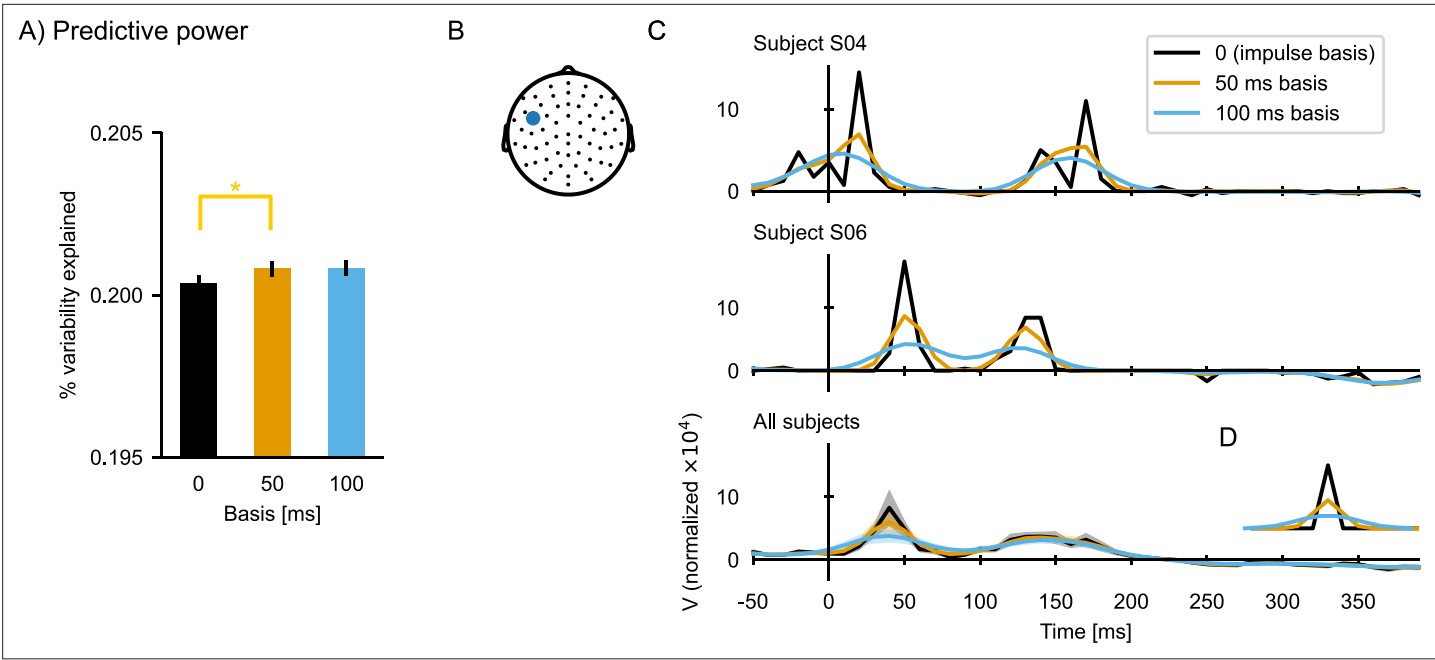

**Figure 10.** The effect of using a basis function for temporal response function (TRF) estimation. (**A**) Predictive power of the TRFs estimated with different basis windows (expressed as a percent of the variability in the EEG data that is explained by the respective TRF model). Error bars indicate the within-subject standard error of the mean, and significance is indicated for pairwise *t*-tests (*df* = 32; *p≤0.05). (**B**) The sensor selected for illustrating the TRFs. (**C**) TRFs for two subject (upper and middle plots) and the average across all subjects (bottom). (**D**) Basis windows. Notice that the impulse basis ('0') is only a single sample wide, but it appears as a triangle in a line plot with the apparent width determined by the sampling rate. Source code: figures/TRF-Basis.py.

with the default impulse basis ('0'). In this case, a 50 ms Hamming window basis leads to significantly better predictive power than the default impulse basis (*Figure 10A*). TRFs are illustrated using a single sensor which exhibits a strong auditory response (*Figure 10B*). *Figure 10C* shows that using a wider basis window leads to markedly smoother TRFs, and increases the overlap of the TRFs between subjects. The basis windows themselves are shown in inset D.

## Acknowledgements

We are grateful for support from the following grants: NSF BCS 1754284, NSF BCS 2043903 & NSF IIS 2207770 (CB); NSF SMA 1734892 (JPK & JZS); NIH R01 DC014085 (JPK & JZS); NIH R01 DC019394 (JZS); FWO SB 1SA0620N (MG); ONR MURI N00014-18-1-2670 (SB & PR); and NIH T32 DC017703 (PG).

## Additional information

### Funding

| Funder | Grant reference number | Author |
|---|---|---|
| National Science Foundation | BCS 1754284 | Christian Brodbeck |
| National Science Foundation | BCS 2043903 | Christian Brodbeck |
| National Science Foundation | IIS 2207770 | Christian Brodbeck |
| National Science Foundation | SMA 1734892 | Joshua P Kulasingham Jonathan Z Simon |

| Funder | Grant reference number | Author |
|---|---|---|
| National Institutes of Health | R01 DC014085 | Joshua P Kulasingham<br>Jonathan Z Simon |
| National Institutes of Health | R01 DC019394 | Jonathan Z Simon |
| Fonds Wetenschappelijk Onderzoek | SB 1SA0620N | Marlies Gillis |
| Office of Naval Research | MURI N00014-18-1-2670 | Shohini Bhattasali<br>Philip Resnik |
| National Institutes of Health | T32 DC017703 | Phoebe Gaston |

The funders had no role in study design, data collection and interpretation, or the decision to submit the work for publication.

### Author contributions

Christian Brodbeck, Conceptualization, Data curation, Software, Formal analysis, Supervision, Validation, Visualization, Methodology, Writing – original draft, Project administration, Writing – review and editing; Proloy Das, Conceptualization, Software, Writing – review and editing; Marlies Gillis, Conceptualization, Formal analysis, Visualization, Writing – original draft, Writing – review and editing; Joshua P Kulasingham, Software, Writing – review and editing; Shohini Bhattasali, Conceptualization, Resources, Writing – review and editing; Phoebe Gaston, Conceptualization, Writing – review and editing; Philip Resnik, Conceptualization, Supervision, Writing – review and editing; Jonathan Z Simon, Conceptualization, Supervision, Funding acquisition, Methodology, Writing – review and editing

### Author ORCIDs

Christian Brodbeck (iD) http://orcid.org/0000-0001-8380-639X
Proloy Das (iD) http://orcid.org/0000-0002-8807-042X
Marlies Gillis (iD) http://orcid.org/0000-0002-3967-2950
Shohini Bhattasali (iD) http://orcid.org/0000-0002-6767-6529
Jonathan Z Simon (iD) http://orcid.org/0000-0003-0858-0698

### Decision letter and Author response

Decision letter https://doi.org/10.7554/eLife.85012.sa1
Author response https://doi.org/10.7554/eLife.85012.sa2

## Additional files

### Supplementary files
• MDAR checklist

### Data availability

The data analyzed here was originally released and can be retrieved from here. For the purpose of this tutorial, the data were restructured and rereleased here. The companion GitHub repository contains code and instructions for replicating all analyses presented in the paper (*Brodbeck et al., 2023*).

The following dataset was generated:

| Author(s) | Year | Dataset title | Dataset URL | Database and Identifier |
|---|---|---|---|---|
| Brodbeck C, Bhattasali S, Das P | 2021 | Data for: Eelbrain: A Python toolkit for time-continuous analysis with temporal response functions | https://doi.org/10.13016/pulf-lndn | Digital Repository at the University of Maryland, 10.13016/pulf-lndn |

The following previously published dataset was used:

| Author(s) | Year | Dataset title | Dataset URL | Database and Identifier |
|---|---|---|---|---|
| Brennan JR | 2020 | EEG Datasets for Naturalistic Listening to "Alice in Wonderland" | https://deepblue.lib.umich.edu/data/concern/data_sets/bg257f92t | Deep Blue Data, 10.7302/Z29C6VNH |

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
