## [Editor Report]

Brodbeck et al. offer a timely and important contribution to how neural signals in response to continuous temporal modulations (as seen in speech and language processing) can be modelled effectively using temporal response functions. They offer a compelling new approach that includes a novel application of a boosting algorithm in addition to an accessible and didactically useful toolbox for analysis. A comparison of boosting and ridge regression via simulation shows the important impact on methods in speech and language neuroscience, as well as in cognitive neuroscience more broadly.

---

## [Decision Letter]

**Decision letter after peer review:**

Thank you for submitting your article "Eelbrain: A Python toolkit for time-continuous analysis with temporal response functions" for consideration by *eLife*. Your article has been reviewed by 2 peer reviewers, and the evaluation has been overseen by a Reviewing Editor and Barbara Shinn-Cunningham as the Senior Editor. The following individual involved in the review of your submission has agreed to reveal their identity: Sophie Slaats (Reviewer #1).

Essential revisions:

1) Both Reviewers helpfully point that more clarity is needed regarding the boosting algorithm; its efficacy needs to be directly compared with existing toolboxes (e.g., Crosse et al., 2016, Weissbart/Reichenbach);

2) Reviewer 2's comments about ridge regression need to be addressed – perhaps comparing boosting and ridge regression with regard to collinearity via simulation could better support the conclusions;

3) Reviewer 2 rightly comments on the importance of accessibility and clarity in the documentation. This needs to be fully addressed in order for the work to have the impact it deserves.

4) Please respond to each of the Reviewer's main concerns, and also take care that important details that impact future users of the toolbox are double-checked (e.g., the broken links need to be fixed).

*Reviewer #1 (Recommendations for the authors):*

This was a great read!

A few questions remain regarding the Boosting algorithm. Firstly, it is not exactly clear how the Sparsity prior is different from the Boosting algorithm, since both appear to favor sparse TRFs (a large number of zeros). Can the boosting algorithm be implemented without a sparsity prior, or is one a logical consequence of the other? It seems like early stopping and assuming an empty mTRF together determine the Sparsity prior. It would be helpful to include this explicitly in section 3.3 (Background: The boosting algorithm), to facilitate the interpretation of (the title of) section 5.2 (Sparsity prior).

Secondly, since the Boosting algorithm sets Eelbrain apart from other TRF implementations, such as the mTRF Toolbox (Crosse et al., 2016) or pyEEG from the Reichenbach lab (https://github.com/Hugo-W/pyEEG), a short comparison of results between two methods would be very informative. Which of the two resulting TRFs is closest to the actual neural response? Why is this the case? Would it be possible that boosting works better in some cases, but not in others?

On page 28 under header 5.4 (Source localization) there is a mention of source localization improving the signal-to-noise ratio of a specific response. While this is undoubtedly true, does this not also lead to the artificial decrease of signal-to-noise ratio for other responses (which might also be relevant, and even modeled)? In other words, is it possible that source localization might lead to an artificial overestimation of the contribution of a given predictor?

The code is easy to use thanks to the wonderful documentation. All the scripts run on the first try. There was only a single error in the scripts as I ran them: in Auditory-TRFs.py, under the header "TRFs", the second cell does not run. The error was the following: [TypeError: 'TTestOneSample' object is not iterable]. All the other cells above it ran to completion. There was a tiny typo under the header "Generate figure" in the same script: the first comment says 'preditive' instead of 'predictive'. Finally, the implementation of the plots yields many warnings if the chosen fonts are not installed (as was the case for me), but this is easy to fix as a user.

The TRFs ran to completion for subjects 1 to 20 – the other ones were omitted due to time constraints. The visual comparison between the newly calculated TRFs and the downloaded ones (for the same subjects, of course) showed minuscule differences in the scalp maps – one or two electrodes were flipped for significance. Unfortunately, the reason for this difference is not clear to me at this point. The other figures were fully identical to the paper.

*Reviewer #2 (Recommendations for the authors):*

1) The main issue I have is that the current toolbox heavily relies on a single way to solve the TRF estimation problem. This is the boosting algorithm. While there are certainly benefits and downsides to any of the solutions, the toolbox forces a method on the user. As the authors explain, the boosting algorithm could come up with a solution in which fully colinear predictors still end up in the model. This would not happen using other algorithms. Interpreting the added benefit of individual predictors to the model could therefore lead to very different conclusions depending on the toolbox you use. It becomes very difficult for any lay audience to compare results from this algorithm to methods implemented in other toolboxes, for example in the mTRF toolbox introduced by Crosse et al. (2016). I am very positive that the authors are open about this and even provide counterintuitive examples of this problem. However, to make the differences clearer for a lay audience I have two improvements here that could be made:

A) Ideally, the toolbox should provide an easy way to implement different ways to solve the TRF estimation. Now, the boosting methods seem difficult to get around in the current implementation (to me). If the authors could provide at least options to change the solution by provide also other solutions themselves. Personally, adding ridge regression would be beneficial as that has been used e.g. in the Crosse et al., (2016) toolbox lot. But if the toolbox is made in a way that custom-made solutions or future solutions can easily be added and compared that would be a great benefit.

B) In the manuscript these methods should be quantitatively compared rather than only describing some counter-intuitive examples of the boosting algorithm. What would ridge regression do for this collinearity problem? I think the manuscript could benefit from a thorough comparison between the different methods.

2) The provided github code together with the manuscript is not very straightforward. While the installation of the toolbox was very smooth, to actually get to the figures provided in the manuscript required going back and forth within all the folders a couple of times. It was not clear to me that first all the code in the predictor/analysis folders must be run before ending up with the results figures etc. Some more instructions on this could have been helpful.

3) If the code in the github should provide the 'easy' way to do the TRF analysis, to me this was not necessarily extremely straightforward. Besides the order issues as described above, getting to the TRF required many steps that remain unexplained. In the code, the main bulk of work is going into making the predictors themselves. This to me makes sense as that is complicated work, however the manuscript itself they give illustrators how to make predictors and provide the code, but it doesn't seem very integral to eelbrain itself to make the predictors in a straightforward manner. So this is left up to the user. If I am a layperson using the toolbox, I, therefore, need to figure out myself outside of the toolbox (potentially based on the code the authors provide, but this does not seem to be an integral part of eelbrain) to make the predictors before I can continue using the code. Of course, a toolbox that helps with the TRF itself is useful, but a user that never has used a TRF before also needs help potentially making the predictors. I guess this is a choice for the authors, which potentially is not clear in the current version of the manuscript. Is the toolbox for solving the TRF problem using the boosting algorithm or is it also intended to provide a means to generate reasonable predictors? If the former is not solving some of the major issues that a lay audience might have if the latter then the manuscript should have way more explanation on how to generate the predictors using eelbrain. This involves not only showing it in the figure but also providing the relevant code, the current descriptions seem insufficient. It seems that the authors focus on making an easy-to-use TRF tool, but provide tools to make predictors but these are not integral to eelbrain. Thus, if the authors aim to provide a tool to go from data to the TRF (including making the predictors), then the manuscript should explain better how this is done and link this to the code. If they do not intend to do this, then they should more clearly separate what eelbrain can and cannot do.

4) Related to issue 3. The use of trftools. In the code for making the predictors, trftools is used a lot. However, when going to the github page of trftools it seems that the authors are not confident about the stability of the code of trftools ("Tools for data analysis with multivariate temporal response functions (mTRFs). This repository mostly contains tools that extend Eelbrain but are not yet stable enough to be included in the main release."). If the code they provide for a publication refers to a toolbox they themselves don't deem stable I find it difficult to judge the value of the toolbox. A layperson might just go along and use the code provided with a published paper.

5) In general, the paper does not provide any tools or directions to use the toolbox. What is the benefit of this toolbox to what is already available? The overall logic of the toolbox and implementation would have been helpful. Now very often the authors refer to function and class types within the toolbox (e.g. NDVar objects etc.), but without any context, this is very difficult to grasp. I understand the manuscript is not the place to provide a full tutorial, but now the manuscript fails to provide an overall logic of the toolbox and how to approach a problem.

6) Regarding this piece of text: "Because the default implementation of the boosting algorithm constructs 1 the kernel from impulses (each element hi,τ is modified independently), this can lead to temporally discontinuous TRFs. In order to derive smoother TRFs, TRFs can be constructed from a basis of smooth window functions instead. In Eelbrain, the basis shape and window size are controlled by the basis and basis_window parameters in the boosting function." Would it be helpful to demonstrate this and also to show what are the default options in the bases and basis_window and why these are chosen in this way? I think it would be useful for a user to know that these are critical choices that are made for them by the toolbox.

---

## [Author Response]

Essential revisions:1) Both Reviewers helpfully point that more clarity is needed regarding the boosting algorithm; its efficacy needs to be directly compared with existing toolboxes (e.g., Crosse et al., 2016, Weissbart/Reichenbach);

We have added more details in the Methods section regarding the boosting algorithm, and we have added a simulation study to demonstrate an essential difference between boosting and ridge regression. For more details see also the responses to the relevant reviewer comments below.

2) Reviewer 2's comments about ridge regression need to be addressed – perhaps comparing boosting and ridge regression with regard to collinearity via simulation could better support the conclusions;

We have added a simulation study that simulates EEG responses to collinear predictor variables, and compares mTRFs reconstructed using boosting and ridge regression. While it is impossible to anticipate all manners of collinearity that users may encounter, the source code associated with the simulation study will provide a template for users to test their own stimuli for issues regarding collinearity.

3) Reviewer 2 rightly comments on the importance of accessibility and clarity in the documentation. This needs to be fully addressed in order for the work to have the impact it deserves.

We have addressed all issues raised regarding the documentation below.

4) Please respond to each of the Reviewer's main concerns, and also take care that important details that impact future users of the toolbox are double-checked (e.g., the broken links need to be fixed).

Please see below for responses.

Reviewer #1 (Recommendations for the authors):This was a great read!

Thank you for the positive evaluation!

A few questions remain regarding the Boosting algorithm. Firstly, it is not exactly clear how the Sparsity prior is different from the Boosting algorithm, since both appear to favor sparse TRFs (a large number of zeros). Can the boosting algorithm be implemented without a sparsity prior, or is one a logical consequence of the other? It seems like early stopping and assuming an empty mTRF together determine the Sparsity prior. It would be helpful to include this explicitly in section 3.3 (Background: The boosting algorithm), to facilitate the interpretation of (the title of) section 5.2 (Sparsity prior).

Thank you for pointing out these questions. We have clarified this in section 3.3 (Background: The boosting algorithm) along with other additions to better explain the algorithm.

Secondly, since the Boosting algorithm sets Eelbrain apart from other TRF implementations, such as the mTRF Toolbox (Crosse et al., 2016) or pyEEG from the Reichenbach lab (https://github.com/Hugo-W/pyEEG), a short comparison of results between two methods would be very informative. Which of the two resulting TRFs is closest to the actual neural response? Why is this the case? Would it be possible that boosting works better in some cases, but not in others?

We have added a simulation study to highlight the key difference between the two algorithms, and the resulting mTRFs’ relationship to the actual neural response (section 3.4.2 Comparison to ridge regression). Which of the two resulting TRFs will be closest to the actual neural response will depend on the metric that measures the ‘closeness’. As we see from the simulation example, boosting mTRFs achieve an excellent true negative rate in expense of biasing the peaks and troughs towards 0 while ridge mTRF closely follows the peaks and troughs, but admits lots of false positives. Now, depending on the hypothesis one is trying to test, in some cases it will be more crucial to have tight control over false positives than other cases. So, ultimately the particular use case and the underlying hypothesis will determine which one of the methods work better.

While a more systematic evaluation and comparison of the two algorithms is beyond the scope of this tutorial, we have also added references to existing work that provides more detailed comparisons between boosting and other reverse correlation algorithms (David et al., 2007; Kulasingham and Simon, 2022). In addition, the source code corresponding to the new simulation example also provides a template for a more in-depth comparison for interested users.

On page 28 under header 5.4 (Source localization) there is a mention of source localization improving the signal-to-noise ratio of a specific response. While this is undoubtedly true, does this not also lead to the artificial decrease of signal-to-noise ratio for other responses (which might also be relevant, and even modeled)? In other words, is it possible that source localization might lead to an artificial overestimation of the contribution of a given predictor?

This is certainly true in individual brain regions. Ideally, the goal of source localization is to isolate the signal originating from the given region while suppressing signals from other regions. In practice, if a certain source region is not modeled (e.g., due to using an inaccurate anatomical model, or only looking at a specific ROI), signals that originated from a region that is not considered might be missed. We have added this consideration to the Source localization section.

The code is easy to use thanks to the wonderful documentation. All the scripts run on the first try. There was only a single error in the scripts as I ran them: in Auditory-TRFs.py, under the header "TRFs", the second cell does not run. The error was the following: [TypeError: 'TTestOneSample' object is not iterable]. All the other cells above it ran to completion. There was a tiny typo under the header "Generate figure" in the same script: the first comment says 'preditive' instead of 'predictive'. Finally, the implementation of the plots yields many warnings if the chosen fonts are not installed (as was the case for me), but this is easy to fix as a user.

Thank you for noting those. The error is fixed with Eelbrain release 0.38.4. We have also changed the font to Arial which is more commonly pre-installed and standard for scientific publications.

The TRFs ran to completion for subjects 1 to 20 – the other ones were omitted due to time constraints. The visual comparison between the newly calculated TRFs and the downloaded ones (for the same subjects, of course) showed minuscule differences in the scalp maps – one or two electrodes were flipped for significance. Unfortunately, the reason for this difference is not clear to me at this point. The other figures were fully identical to the paper.

The miniscule differences in the TRFs are likely due to the optimization of the boosting implementation in Eelbrain 0.38. This implementation change may have led to slight changes of results due to numerical inaccuracy issues (performing operations in a different sequence, etc.). We have now re-estimate all TRFs with the optimized algorithm and added them to the DRUM repository that hosts the EEG data.

Reviewer #2 (Recommendations for the authors):1) The main issue I have is that the current toolbox heavily relies on a single way to solve the TRF estimation problem. This is the boosting algorithm. While there are certainly benefits and downsides to any of the solutions, the toolbox forces a method on the user. As the authors explain, the boosting algorithm could come up with a solution in which fully colinear predictors still end up in the model. This would not happen using other algorithms. Interpreting the added benefit of individual predictors to the model could therefore lead to very different conclusions depending on the toolbox you use. It becomes very difficult for any lay audience to compare results from this algorithm to methods implemented in other toolboxes, for example in the mTRF toolbox introduced by Crosse et al. (2016). I am very positive that the authors are open about this and even provide counterintuitive examples of this problem. However, to make the differences clearer for a lay audience I have two improvements here that could be made:A) Ideally, the toolbox should provide an easy way to implement different ways to solve the TRF estimation. Now, the boosting methods seem difficult to get around in the current implementation (to me). If the authors could provide at least options to change the solution by provide also other solutions themselves. Personally, adding ridge regression would be beneficial as that has been used e.g. in the Crosse et al., (2016) toolbox lot. But if the toolbox is made in a way that custom-made solutions or future solutions can easily be added and compared that would be a great benefit.

We completely agree that it would be desirable to have an option of different estimation algorithms in the same package. Eelbrain’s boosting implementation is open and highly modular, and we would welcome community contributions for other algorithms. While we do not want to delay publication of this tutorial for such implementations, we have now added a simulation study that uses boosting and ridge regression alongside each other, which provides a template for users that want to compare the two algorithms further (section 3.4.2 Comparison to ridge regression).

B) In the manuscript these methods should be quantitatively compared rather than only describing some counter-intuitive examples of the boosting algorithm. What would ridge regression do for this collinearity problem? I think the manuscript could benefit from a thorough comparison between the different methods.

We have now added a simulation study to compare the boosting algorithm and ridge regression in presence of multiple collinear predictor variables. In the spirit of open source we chose to use the Python pyEEG toolbox from the Reichenbach lab (https://github.com/Hugo-W/pyEEG), over the MATLAB mTRF toolbox. As evident from the simulation example, both the boosting mTRF and ridge mTRF capture the dominant components of the ground truth mTRFs. But, the distinguishing factor between them is that the ridge mTRF follows the dominant portions of the true mTRF at the expense of numerous false positive discoveries while the boosting mTRF allows a small bias towards 0 in the peaks and troughs of the mTRFs to maintain a tight control over false positives. A detailed, systematic comparison between these two methods is out of scope of this work. However, we pointed the readers to a previous work that performed an extensive comparison between boosting and other reverse correlation algorithms (David et al., 2007; Kulasingham and Simon, 2022). In addition, we have also added the source code for this simulation as a platform for enthusiastic readers to carry out more in-depth comparisons.

2) The provided github code together with the manuscript is not very straightforward. While the installation of the toolbox was very smooth, to actually get to the figures provided in the manuscript required going back and forth within all the folders a couple of times. It was not clear to me that first all the code in the predictor/analysis folders must be run before ending up with the results figures etc. Some more instructions on this could have been helpful.

Thank you for pointing out this lack of clarity. We have added corresponding instructions to the README.md file of the Alice repository.

3) If the code in the github should provide the 'easy' way to do the TRF analysis, to me this was not necessarily extremely straightforward. Besides the order issues as described above, getting to the TRF required many steps that remain unexplained. In the code, the main bulk of work is going into making the predictors themselves. This to me makes sense as that is complicated work, however the manuscript itself they give illustrators how to make predictors and provide the code, but it doesn't seem very integral to eelbrain itself to make the predictors in a straightforward manner. So this is left up to the user. If I am a layperson using the toolbox, I, therefore, need to figure out myself outside of the toolbox (potentially based on the code the authors provide, but this does not seem to be an integral part of eelbrain) to make the predictors before I can continue using the code. Of course, a toolbox that helps with the TRF itself is useful, but a user that never has used a TRF before also needs help potentially making the predictors. I guess this is a choice for the authors, which potentially is not clear in the current version of the manuscript. Is the toolbox for solving the TRF problem using the boosting algorithm or is it also intended to provide a means to generate reasonable predictors? If the former is not solving some of the major issues that a lay audience might have if the latter then the manuscript should have way more explanation on how to generate the predictors using eelbrain. This involves not only showing it in the figure but also providing the relevant code, the current descriptions seem insufficient. It seems that the authors focus on making an easy-to-use TRF tool, but provide tools to make predictors but these are not integral to eelbrain. Thus, if the authors aim to provide a tool to go from data to the TRF (including making the predictors), then the manuscript should explain better how this is done and link this to the code. If they do not intend to do this, then they should more clearly separate what eelbrain can and cannot do.

Thank you for drawing attention to this shortcoming. We have added line-by-line comments to the scripts for generating the predictor variables, so that there should be no more unexplained steps required. In order to show users where to start to learn more about functionality allowing them to create new predictors, we have also added explicit pointers to the relevant functionality in Eelbrain in the Predictor variables subsection of the Methods (from there, users will be able to use the online documentation of those functions to find further help).

4) Related to issue 3. The use of trftools. In the code for making the predictors, trftools is used a lot. However, when going to the github page of trftools it seems that the authors are not confident about the stability of the code of trftools ("Tools for data analysis with multivariate temporal response functions (mTRFs). This repository mostly contains tools that extend Eelbrain but are not yet stable enough to be included in the main release."). If the code they provide for a publication refers to a toolbox they themselves don't deem stable I find it difficult to judge the value of the toolbox. A layperson might just go along and use the code provided with a published paper.

The functionality from TRF-Tools that was required for this analysis has been moved to Eelbrain and TRF-Tools is no longer referenced from the Alice repository.

5) In general, the paper does not provide any tools or directions to use the toolbox. What is the benefit of this toolbox to what is already available? The overall logic of the toolbox and implementation would have been helpful. Now very often the authors refer to function and class types within the toolbox (e.g. NDVar objects etc.), but without any context, this is very difficult to grasp. I understand the manuscript is not the place to provide a full tutorial, but now the manuscript fails to provide an overall logic of the toolbox and how to approach a problem.

Thank you for pointing out this omission. We added a section describing the general structure of the toolbox at the beginning of the Methods section, under the heading “Overall architecture of the Eelbrain toolbox”

6) Regarding this piece of text: "Because the default implementation of the boosting algorithm constructs 1 the kernel from impulses (each element hi,τ is modified independently), this can lead to temporally discontinuous TRFs. In order to derive smoother TRFs, TRFs can be constructed from a basis of smooth window functions instead. In Eelbrain, the basis shape and window size are controlled by the basis and basis_window parameters in the boosting function." Would it be helpful to demonstrate this and also to show what are the default options in the bases and basis_window and why these are chosen in this way? I think it would be useful for a user to know that these are critical choices that are made for them by the toolbox.

We have added a discussion of the effect of using different basis windows at the end of the Methods section.